# *qsmR* encoding an IclR-family transcriptional factor is a core pathogenic determinant of *Burkholderia glumae* beyond the acyl-homoserine lactone-mediated quorum-sensing system

Tiago De Paula Lelis[¤][☉], Jobelle Bruno[☉], Jonas Padilla, Inderjit Barphagha, John Ontoy, Jong Hyun Ham[iD]*

Department of Plant Pathology and Crop Physiology, Louisiana State University Agricultural Center, Baton Rouge, Louisiana, United States of America

☉ These authors contributed equally to this work.
¤ Current Address: Invaio Sciences, Cambridge, Massachusetts, United States of America
* jham@agcenter.lsu.edu

**Data Availability Statement:** All relevant data are within the manuscript and its Supporting

## Abstract

The plant pathogenic bacterium *Burkholderia glumae* causes bacterial panicle blight (BPB) in rice-growing areas worldwide. It has been widely accepted that an acyl-homoserine lactone (AHL)-type quorum sensing (QS) system encoded by *tofI* and *tofR* genes (TofIR QS) is a key regulatory mechanism underlying the bacterial pathogenesis of *B. glumae*. In addition, *qsmR*, which encodes an IclR-family regulatory protein, has been considered an important part of TofIR QS. However, the present study with three strains of *B. glumae* representing different pathogenic strains revealed that this currently accepted paradigm should be modified. We characterized the regulatory function of TofIR QS and *qsmR* in three different strains of *B. glumae*, 336gr-1 (virulent), 411gr-6 (hypervirulent) and 257sh-1 (avirulent). In 336gr-1, both TofIR QS and *qsmR* were critical for the pathogenesis, being consistent with previous studies. However, in the hypervirulent strain 411gr-6, TofIR QS only partially contributes to the virulence, whereas *qsmR* was critical for pathogenesis like in 336gr-1. Furthermore, we found that a single nucleotide polymorphism causing T50K substitution in the *qsmR* coding sequence was the cause of the non-pathogenic trait of the naturally avirulent strain 257sh-1. Subsequent analyses of gene expression and transcriptome revealed that TofIR QS is partially controlled by *qsmR* at the transcriptional level in both virulent strains. Further genetic tests of additional *B. glumae* strains showed that 11 out of 20 virulent strains retained the ability to produce toxoflavin even after removing the *tofI/tofM/tofR* QS gene cluster like 411gr-6. In contrast, all the virulent strains tested lost the ability to produce toxoflavin almost completely upon deletion of the *qsmR* gene. Taking these results together, *qsmR*, rather than TofIR QS, is a master regulator that determines the pathogenic trait of *B. glumae* thus a more appropriate pathogen target for successful management of BPB.

information files. The raw RNA-seq data used in this study were deposited to the NCBI database with the BioProject Accession ID, PRJNA1110263 (SRA IDs, SRR28995946 - SRR28995963).

**Funding:** This study was supported by the USDA National Institute of Food and Agriculture (Grants, 2022-67013-36140 and 2023-68012-39002; Hatch, CT0477) (JB, JP, IB, JO, JHH), the Louisiana Rice Research Board (GR-00006499) (JB, IB), and CNPq (Conselho Nacional de Desenvolvimento Científico e Tecnológico) (TL). The funders had no role in study design, data collection and analysis, decision to publish, or preparation of the manuscript.

**Competing interests:** The authors have declared that no competing interests exist.

## Author summary

Bacterial cell-to-cell communication, called quorum-sensing, systems mediated by homoserine lactone (HSL)-type signaling molecules play pivotal roles in the virulence-related functions of diverse plant and animal pathogenic Gram-negative bacteria. *Burkholderia glumae* is the chief causal agent for bacterial panicle blight, a devastating rice disease worldwide. In this pathogen, the HSL-type quorum-sensing system dependent on the *tofI* and *tofR* genes had long been thought to be the core regulatory/signaling system essential for virulence. However, we discovered in this study that a highly virulent strain of *B. glumae*, 411gr-6, retained its virulence even after the *tofI/tofR*-dependent quorum-sensing function was disabled. We also found that more than a half of the natural strains of this pathogen species tested in this study also exhibited their virulence function in the absence of the *tofI/tofR*-dependent quorum-sensing system. Then we characterized the *qsmR* gene as an essential regulatory/signaling element for the bacterial pathogenesis regardless of the genetic variations among the strains of *B. glumae*. This study indicates that the *qsmR* gene, rather than the *tofI/tofR*-dependent quorum-sensing system, should be considered as an essential master regulatory/signaling factor for the virulence of *B. glumae* and, thus, a promising target for suppressing bacterial panicle blight.

## Introduction

*Burkholderia glumae* is a plant pathogenic bacterium that causes bacterial panicle blight (BPB) in rice. Since first reported in East Asia, outbreaks of BPB have been reported globally, occasionally resulting in severe yield losses [1,2]. Even though the disease cycle and epidemiology of BPB have not been characterized well enough, it is generally accepted that *B. glumae*, the major causal agent of BPB, is a seed-borne pathogen, and toxoflavin, an azapteridine phytotoxin, is the most important virulence factor of the pathogen [3]. Because *B. glumae* is also pathogenic to other important crops and hard to control with available chemical measures for bacterial pathogens, such as antibiotics and copper compounds, it is imperative to understand the signaling and regulatory system governing the pathogenic function of this pathogen and to identify its key elements of pathogenesis for developing target-specific disease management strategy against this pathogen [4,5].

It has been widely accepted that major virulence functions of *B. glumae*, especially toxoflavin production and flagellum-mediated motility, are regulated by an acyl-homoserine lactone (AHL)-type quorum sensing (QS) system encoded by *tofI* and *tofR* genes (TofIR QS) [6–9]. In the AHL-type QS system of *B. glumae*, *tofI*, a *luxI* homolog, is responsible for synthesizing *N*-octanoyl homoserine lactone (C8-HSL), *tofR* encodes a LuxR-family transcriptional regulator activated by C8-HSL, and the TofR-C8-HSL complex leads to the expression of *toxR* and *toxJ* for toxoflavin biosynthesis and transport [6] and of *flhD* and *flhC* for flagella biosynthesis [8]. In addition, *tofM* was identified in the intergenic region between *tofI* and *tofR* in our previous study as a modulator of TofIR QS [10].

Besides TofIR QS, *qsmR*, which encodes an IclR-type transcriptional regulator, also plays a pivotal regulatory role in the virulence of *B. glumae*. This gene was first reported as a regulatory component for the flagella biosynthesis and virulence of *B. glumae* BGR1 [8]. In our previous genetic studies with another virulent strain of *B. glumae*, 336gr-1, *qsmR* was essential for the regulation of toxoflavin and extracellular protease [11] and was

identified as one of the genetic elements for the TofI-independent production of toxoflavin [7]. Furthermore, additional studies on *qsmR* elucidated the pivotal regulatory function of this gene in the bacterial metabolism to produce oxalate, which is a highly critical cooperative behavior of *Burkholderia* spp. for the survival in the stationary phase of bacterial population [12,13].

Although TofIR QS has been known as the central regulatory system for virulence-related functions in *B. glumae*, variations in the mutant phenotypes for individual *tofI* and *tofR* genes have also been reported. In *B. glumae* BGR1, null mutation of *tofI* or *tofR* led to the complete abolishment of toxoflavin production [6]. However, our previous studies conducted with *B. glumae* 336gr-1 showed that the single-gene deletion mutants for *tofI* and *tofR* (i.e., Δ*tofI* and Δ*tofR* derivatives of 336gr-1, respectively) retained the ability to produce toxoflavin in a solid medium condition (i.e., LB agar medium), while the deletion of the whole QS gene cluster (*tofI/tofM/tofR*) led to the complete loss of toxoflavin production in both liquid and solid medium conditions [10]. Furthermore, a similar study by another research group reported that deletion of the *tofI* gene in 14 strains of *B. glumae* isolated from Japan revealed variations of their *tofI⁻* phenotype in toxoflavin production, where most of the strains (11 out of 14) produced toxoflavin in both LB agar and LB broth conditions, two strains showed the phenotypes like BGR1 (toxoflavin negative in both liquid and solid medium conditions), and one strain phenotypically behaved like 336gr-1 (toxoflavin negative in the liquid medium but positive in the solid medium consition) [14]. These previous studies suggested that the contribution of individual QS genes, *tofI* and *tofR*, to the virulence of *B. glumae* varies among different strains, although TofIR QS conferred by the collective function of the *tofI/tofM/tofR* gene cluster is required for bacterial pathogenesis.

Meanwhile, we previously reported a wide range of diversity among field strains of *B. glumae* in terms of the virulence in the host (rice) and the surrogate host (onion bulb) and the pigment production on CPG agar medium [15]. In that study, we identified hypervirulent, virulent, hypo-virulent, and naturally avirulent strains. Further, we sequenced the whole genomes of four strains representing different phenotypes in virulence and pigment production on CPG agar medium: 336gr-1 (virulent and non-pigment-producing), 411gr-6 (highly virulent and pigment-producing), 957856-41-c (hypo-virulent and non-pigment-producing), and 257sh-1 (avirulent and pigment-producing) [16].

In this study, we performed genetic and genomic studies on three strains of *B. glumae*, 336gr-1, 411gr-6, and 257sh-1, to characterize the genetic elements and regulatory systems that determine their differential pathogenic traits. *B. glumae* 336gr-1 is a virulent, non-pigmented strain that produces the virulence factors toxoflavin and the PrtA extracellular protease, which represent the most important and a moderately important virulence factors of *B. glumae*, respectively [10,11,17]. *B. glumae* 411gr-6 is a highly virulent strain that produces pigments on CPG medium, which have antimicrobial activities [15,18]. *B. glumae* 257sh-1 is naturally avirulent but shows the pigmentation phenotype on CPG medium, like *B. glumae* 411gr-6 [15]. From this study, we discovered that TofIR QS makes only a partial contribution to the pathogenesis of *B. glumae* in some strains, including 411gr-6, while *qsmR* functions as an absolute regulatory determinant of pathogenicity in all the 20 strains of *B. glumae* tested regardless of the phenotypic diversity among strains. The results obtained from this study support a new model that defines the differential roles of TofIR QS and the QsmR transcriptional regulator for the pathogenic behaviors of this bacterial pathogen.

## Results

### Comparative analysis of three *B. glumae* genomes that represent differential phenotypic traits in virulence and pigmentation

The whole genome sequence information of the *B. glumae* strains 336gr-1 (virulent, non-pigmenting), 257sh-1 (avirulent, pigmenting), and 411gr-6 (hyper-virulent, pigmenting) was announced previously [16]. Their phenotypic traits are presented in Fig 1. Each strain contains two chromosomes and two plasmids except for the 336gr-1 strain which has three plasmids. The strains also have genome sizes that range from 6,691,685 to 6,879,338 bp (Table A in S1 Text). In this study, we conducted a phylogenetic analysis to investigate the relatedness of the three *B. glumae* strains along with eight other strains based on the single-copy genes from each strain, which revealed that *B. glumae* strains 257sh-1 and 411gr-6 clustered together, while 336gr-1 was in a different clade along with other strains from different geographic origins, such as BGR1 from Korea (Fig A in S1 Text). This is consistent with our previous study, which reported the closer genetic relatedness between 257sh-1 and 411gr-6 based on rep-PCR and their pigmentation phenotypes on CPG medium [15]. The pairwise comparison among the *B. glumae* strains 336gr-1, 411gr-6, and 257sh-1 also revealed that the strains 257sh-1 and 411gr-1 share the highest average nucleotide identity (ANI) among other pairwise comparisons (Table B in S1 Text).

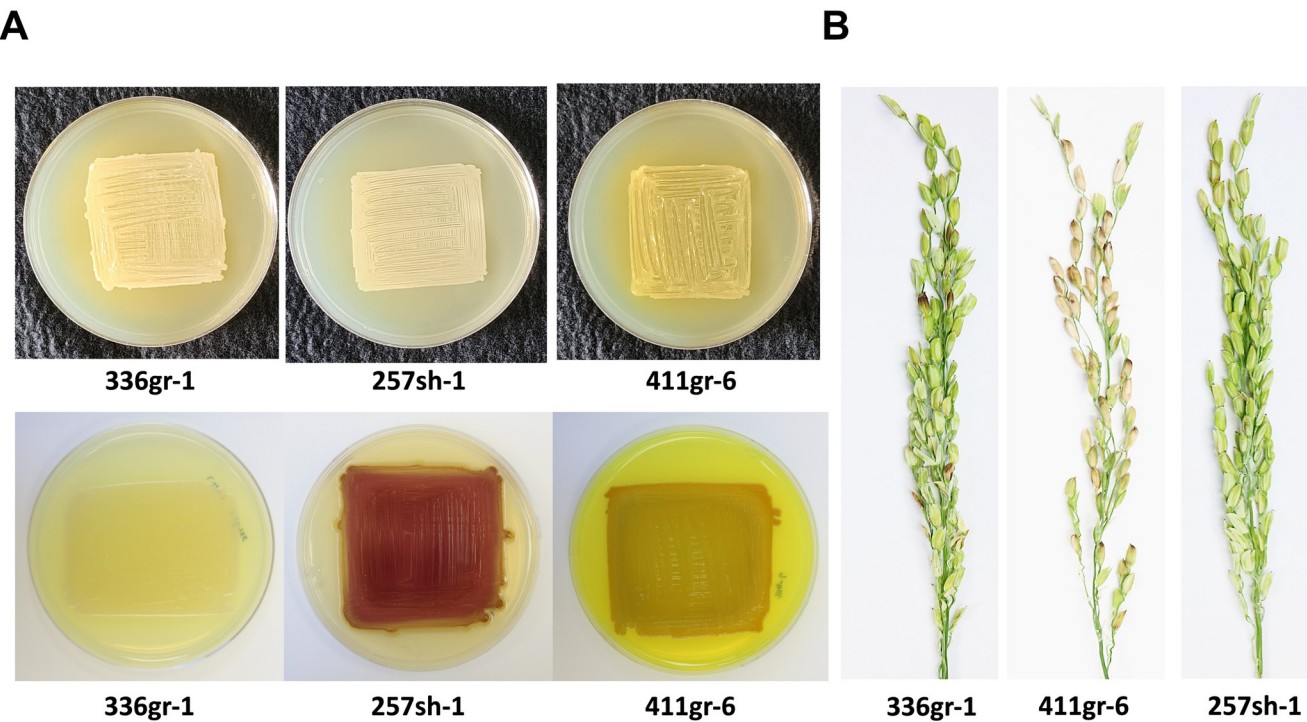

**Fig 1. The phenotypes of the three *Burkholderia glumae* wild type strains, 257sh-1, 336gr-1, and 411gr-6.** (A) The differential colony morphology of *B. glumae* strains on the LB agar medium (upper panel) and the CPG agar medium (lower panel). (B) The differential virulence phenotypes of *B. glumae* strains on rice panicles. *The B. glumae* strains on LB agar and CPG agar were photographed 24 h and 48 h after incubation at 37°C, respectively, for panel A. The rice panicle pictures in panel B were taken 11 days after inoculation.

## Comparative analysis of three *B. glumae* genomes focusing on genes for bacterial pathogenesis

We further characterized the three genomes of *B. glumae* to identify possible genetic elements associated with their pathogenic traits, especially for the natural avirulence of 257sh-1. With the hypothesis that the different virulence phenotypes of the three strains are due to mutation or sequence variation in a gene or genes critical for pathogenic function, the comparative analysis was primarily focused on genes for known and potential virulence factors as well as their regulatory and secretory components. Total of 85 genes were selected to identify amino acid sequence variations specific to the avirulent strain 257sh-1 based on their known or potential virulence-related functions, which included toxoflavin biosynthesis and transport, flagellum-mediated motility, lipase (*lipA*), extracellular metalloprotease (*prtA*), TofIR QS (*tofI/tofM/tofR*), the PidS/PidR two component regulatory system (*pidS/pidR*), transcriptional regulators of virulence factors (*qsmR*, *toxR*, *toxJ*, *flhDC*, *tepR*, *cidR*, *marR*), and protein secretion systems (type II, III, and VI) (Table C in S1 Text). Pairwise comparisons of the 85 genes revealed that 29 genes had amino acid sequence variations in at least one of the three strains, among which only *qsmR* had a variation specific to the avirulent strain 257sh-1 (Table C in S1 Text).

## Deletion of the *tofI/tofM/tofR* cluster in the hypervirulent strain of *B. glumae*, 411gr-6, did not abolish toxoflavin production and extracellular protease activity unlike the virulent strain 336gr-1

In the previous studies with the virulent strains 336gr-1 and BGR1, the TofIR QS system was essential for the virulence functions of *B. glumae*, including toxoflavin production [10] [6] and extracellular protease activity [11]. In this study, we generated a *ΔtofI-tofR* (deletion of the *tofI/tofM*/tofR cluster) derivative of 411gr-6, 411ΔtofI-R, to examine if TofIR QS also exerts the same regulatory function for virulence in this hypervirulent strain (Fig B in S1 Text). The culture extract from both wild types, 336gr-1 and 411gr-6, induced violacein production by the biosensor *Chromobacterium violaceum* CV026 [19], while that from their *ΔtofI-tofR* derivatives, 336ΔtofI-R and 411ΔtofI-R, did not induce observable violacein, indicating their loss of function in AHL synthesis (Fig C in S1 Text).

Remarkably, the *ΔtofI-tofR* derivative of 411gr-6 retained the ability to produce toxoflavin, which was visualized as the yellow color of the bacterial colony on the LB agar medium and quantified after chloroform extraction (Fig 2A and 2B). Although this mutant deficient in TofIR QS produced toxoflavin at a lower level compared to its parent strain 411gr-6, its capacity to produce toxin was comparable to the less virulent strain 336gr-1 (Fig 2A and 2B). In contrast, the *ΔtofI-tofR* derivative of 336gr-1 almost completely lost its ability to produce toxoflavin (Fig 2A and 2B). A similar pattern was observed with the same set of *B. glumae* strains for the phenotypes in extracellular protease activity, which was visualized on the NA agar medium containing 1% skim milk and quantified using azocasein (Fig 2C and 2D).

## The *ΔtofI-tofR* derivative of 411gr-6 retained the ability to cause symptoms on rice panicles

It was reported previously that TofIR QS regulates the expression of major virulence factors in *B. glumae*, and disruption of this system led to a drastic reduction of disease severity caused by *B. glumae* BGR1 and 336gr-1 on rice panicles [6,10]. Since 411ΔtofI-R retained a high level of toxoflavin production and extracellular protease, we hypothesized that this mutant would be virulent to rice plants. To evaluate the virulence of 411ΔtofI-R, rice panicles were inoculated

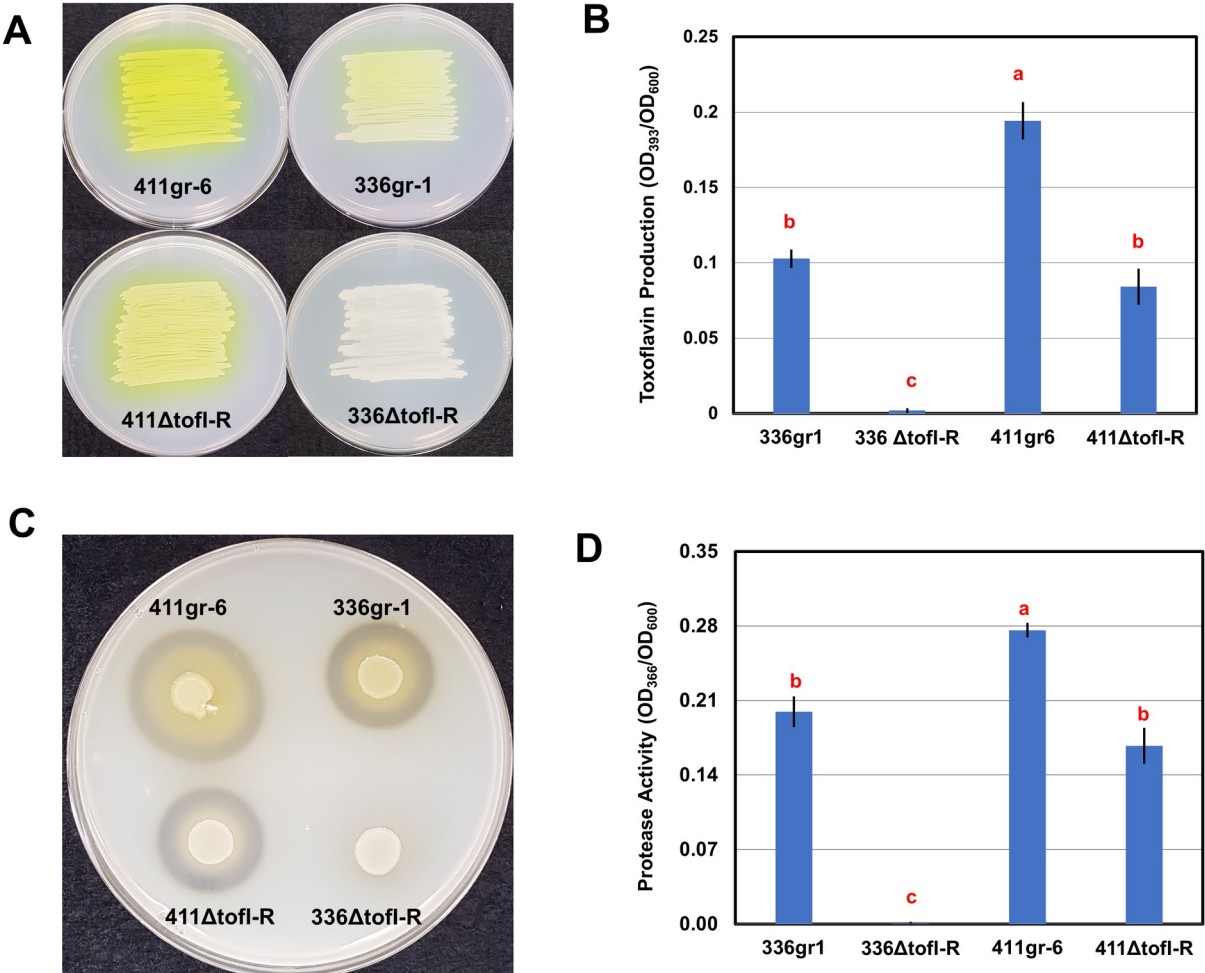

**Fig 2. The differential functions of TofIR QS in 336gr-1 and 411gr-6 in toxoflavin production and extracellular protease activity.** A) The toxoflavin production phenotypes of the *B. glumae* strains on the LB agar. Bacterial cells were inoculated via streaking method onto LB agar media. Toxoflavin production is shown as yellow pigment diffused on solid media. The photo was taken 24 h after inoculation. B) Quantitative data of toxoflavin produced by the *B. glumae* strains. *B. glumae* strains were grown overnight in LB broth, and toxoflavin was extracted with chloroform. C) The extracellular protease phenotypes of the *B. glumae* strains on the indicator medium (NA agar with 1% skim milk). Bacterial strains were inoculated onto Nutrient Agar (NA) plate supplemented with 1% skim milk. The clear zones surrounding bacterial colonies indicate proteolytic activity. The photo was taken 48 h after inoculation. D) The extracellular protease activities of the *B. glumae* strains quantified using azocasein. *B. glumae* strains were grown overnight in LB broth and proteolytic activity of the cell-free supernatant was quantified using azocasein as the protease substrate. Error bars of each graph represent the standard deviation of three replications, and the letters on data columns indicate statistically significant differences at $P < 0.05$ based on Tukey's post-hoc test. Three independent experiments yielded similar results.

with the strain along with its parent 411gr-6 as well as 336gr-1 and its QS-deficient derivative 336ΔtofI-R. In the greenhouse tests for virulence, both wild type strains 336gr-1 and 411gr-6 were able to induce a high level of disease severity in the rice panicles with disease scores above 5.5 and 7.5 in a 0–9 scale disease severity index, respectively (Fig 3A and 3B). The QS mutants, 336ΔtofI-R and 411ΔtofI-R, were substantially less virulent than their parent strains but still able to cause disease, showing ca. 47% and 31% reduction of disease severity, respectively, compared to their parent wild types (Fig 3A and 3B). Remarkably, 411ΔtofI-R exhibited a similar level of virulence to 336gr-1 (Fig 3A and 3B).

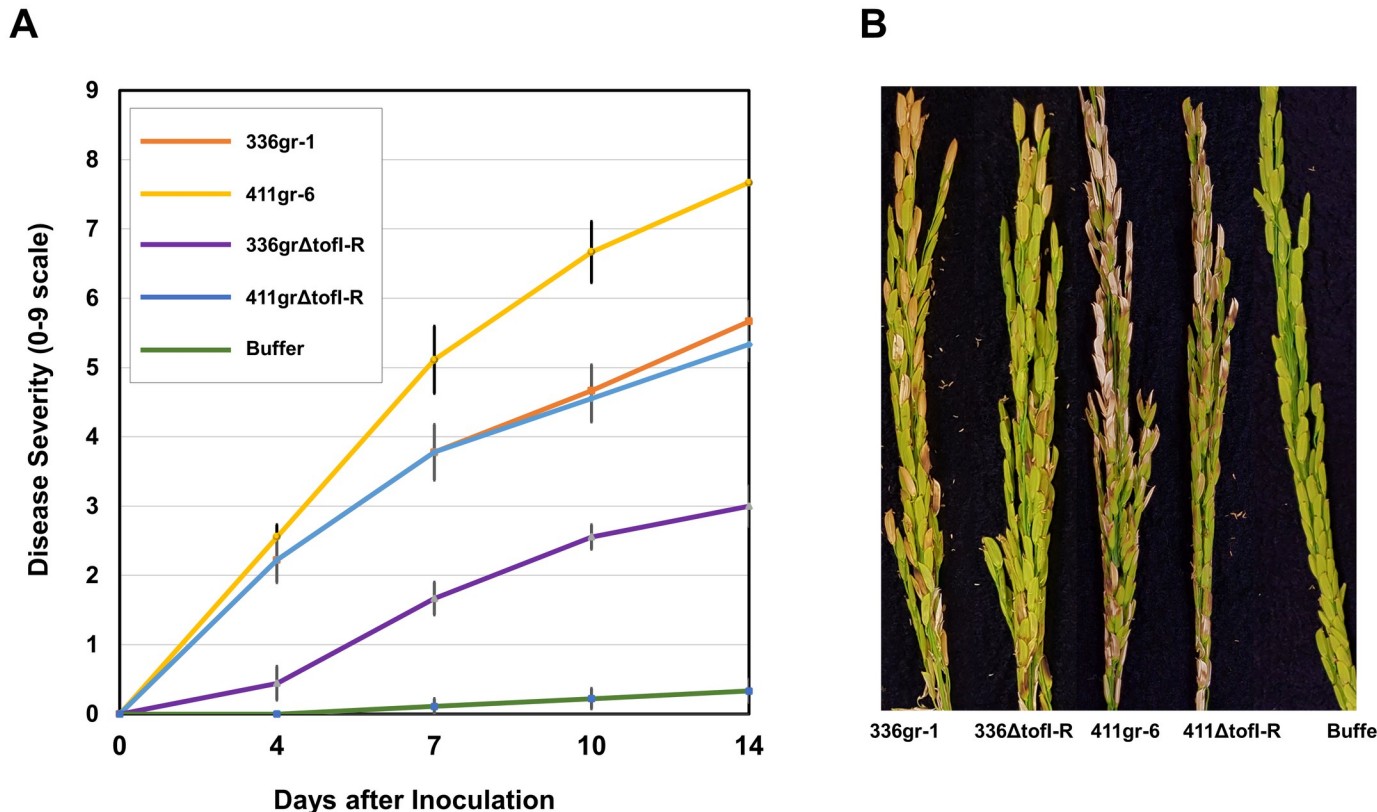

**Fig 3. The differential phenotypes of 336gr-1 and 411gr-6 and their TofIR QS-deficient (*ΔtofI-R*) derivatives in the symptom development on rice panicles.**
A) Disease progress curves caused by *B. glumae* strains. Disease severity was scored 5 times in a 14-days period after inoculation, using a 0–9 scale, in which 0 indicated no symptoms and 9 indicated more than 80% discolored panicles. Error bar represents the standard deviation of five replications. B) Disease symptoms on rice panicles caused by the *B. glumae* strains. The photo was taken 14 days after inoculation. Greenhouse experiments were repeated three times, yielding similar results.

## The gene expression patterns of toxoflavin-related genes determined by qRT-PCR were consistent with the observed toxoflavin production phenotypes

To validate the observed phenotypes of TofIR QS-deficient mutants in toxoflavin production, transcription patterns of the genes involved in toxoflavin production were examined using the quantitative reverse transcription (qRT)-PCR technique, which were *toxA*, *toxJ*, *toxR*, and *qsmR*. The *toxA* gene is the first gene of the *toxABCDE* operon, which encodes enzymes for biosynthesis of toxoflavin, and the *toxJ* and *toxR* genes are regulatory genes that directly control the genes for biosynthesis and transport of the phytotoxin [6]. The *qsmR* gene was first identified as a regulatory gene that controls the flagellum biogenesis, as well as the toxoflavin production, in *B. glumae* [8].

In the qRT-PCR tests of this study, both regulatory genes for toxoflavin production, *toxJ* and *toxR*, were almost completely repressed in transcription in both 336gr-1 and 411gr-6 via disruption of TofIR QS function (Fig 4). However, transcription of *toxA*, the first gene of the *toxABCDE* operon for toxoflavin biosynthesis, was only partially (but significantly) impaired in 411gr-6 by the disruption of TofIR QS, showing its transcription level comparable to 336gr-1 (Fig 4). These results were congruent with the phenotypes observed in toxoflavin production (Fig 2). They indicate that the expression of the *toxABCDE* operon, which is responsible for the

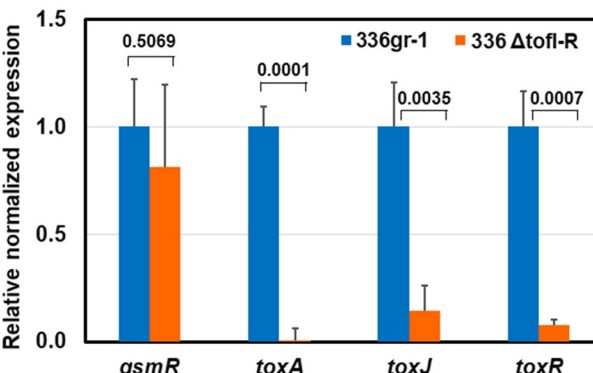
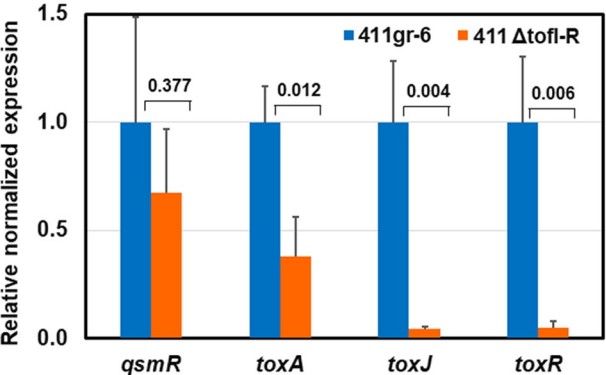

**Fig 4. The relative transcription levels of toxoflavin-related genes determined by qRT-PCR in the *ΔtofI-R* derivatives of 336gr-1(336ΔtofI-R) and 411gr-6 (411ΔtofI-R).** Fold change of each gene in qRT-PCR was calculated with the $2^{-\Delta\Delta Ct}$ method, and expression was normalized using the reference genes, *gyrA* and 16S rRNA. Experiment was repeated twice, and error bars represent the standard deviation of experiments with three replications. The numbers on the top of each pair of columns indicate the p-values based on two-tail t-test.

biosynthesis of toxoflavin, is not solely dependent on TofIR QS in 411gr-6 (Fig 4). Interestingly, *qsmR* was not significantly affected by TofIR QS in both 336gr-1 and 411gr-6 (Fig 4), although this gene was first reported to be dependent on TofIR QS in the Korean strain BGR1[8].

## *qsmR* was required for toxoflavin production and extracellular protease activity in both 336gr-1 and 411gr-6

Since the expression level of *qsmR* was not significantly affected by TofIR QS in both 336gr-1 and 411gr-6 (Fig 4), we generated a *qsmR*-deficient mutant of each strain to investigate its regulatory function in toxoflavin production and extracellular protease activity and further its functional relationship with TofIR QS in both strains. Remarkably, deletion of *qsmR* abolished toxoflavin production and extracellular protease activity in both 336gr-1 and 411gr-6 (Fig 5). In the same experiment, the TofIR QS-deficient derivatives (*ΔtofI-R*) of both strains showed the same pattern as Fig 2, which showed only partial reduction of toxoflavin production and extracellular protease activity by *ΔtofI-R* mutation in 411gr-6 but almost complete loss of functions in 336gr-1 by the same mutation (Fig 5). These results indicated that *qsmR* exerted a master regulatory role for toxoflavin production and extracellular protease activity in both strains.

## Deletion of *qsmR* in 336gr-1 and 411gr-6 led to the abolishment of disease symptoms in rice seedlings and panicles

Because the virulence-related functions, toxoflavin production and extracellular protease activity, were not detected in the *ΔqsmR* derivatives of both 336gr-1 and 411gr-6, we also investigated the virulence phenotype of these mutants in rice plants to determine how the virulence-related phenotypes observed *in vitro* reflect the pathogenic ability of the pathogen *in vivo*. Rice plants were inoculated with each strain of *B. glumae* at the seedling stage. Symptom development on rice sheaths was assessed 14 days after inoculation and the bacterial population in each inoculated plant was also determined by real-time-PCR using the same plant samples.

Symptoms of wild type strains (336gr-1 and 411gr-6) initially appeared as browning around the inoculation sites. They expanded to a long lesion extending to most rice sheath (Fig 6A). 336ΔtofI-R made small legions limited to the surrounding area of the inoculation sites, while 411ΔtofI-R showed longer lesions extended to other parts of the rice sheath like its parent strain 411gr-6 (Fig 6A). In contrast, *qsmR* mutants of 336gr-1 and 411gr-6, 336ΔqsmR and

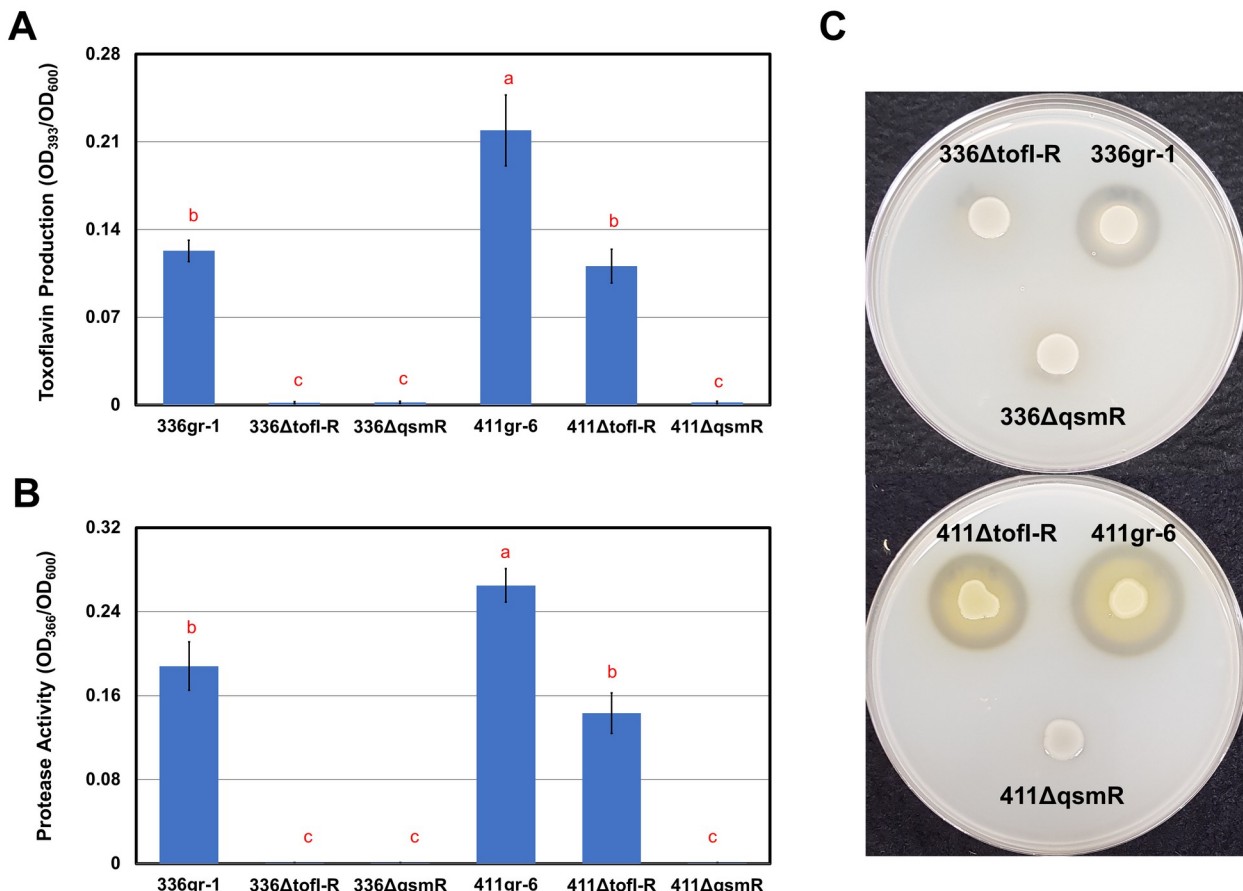

**Fig 5. The phenotypes of *qsmR* deletion mutation in toxoflavin production and extracellular protease activity.** A) The toxoflavin production in LB broth by the *B. glumae* strains. B) The proteolytic activity of the cell-free supernatant from each cell culture of the *B. glumae* strains. C) The extracellular protease phenotypes of the *B. glumae* strains on the NA agar supplemented with 1% skim milk. Experiments were repeated three times and error bars represent the standard deviation of three biological replications. Bars with different letters indicate statistically significant differences among the data curves at $P < 0.05$ based on Tukey's post-hoc test.

411ΔqsmR, respectively, showed almost no observable symptom development in rice seedlings in this experimental condition (Fig 6A). Consistent with the observed pattern of symptom development, the population level of 411gr-6 was the highest among all strains tested, and 336gr-1 and 411ΔtofI-R showed a similar population level to each other (Fig 6B). However, basal levels of bacterial population were still detected for the *qsmR*-deficient mutants of both 336gr-1 and 411gr-6 (Fig 6B). The wild type and mutant strains showed a similar pattern in their symptom development ability in rice panicles (Fig 6C and 6D).

### Deletion of *qsmR* affected the expression of *tofI* and *tofR* differentially between 336gr-1 and 411gr-6

To investigate the regulatory function of *qsmR* on TofIR QS, we conducted qRT-PCR experiments and measured the relative transcription levels of *tofI* and *tofR* in the *ΔqsmR* backgrounds compared to the corresponding wild type strains. The *ΔtofI-R* derivatives were also included, as well as the expression of *toxA* was also measured to affirm the reliability of the qRT-PCR data by determining the reproducibility of those for the *ΔtofI-R* derivatives shown in Fig 4. As presented in Fig 7A, the transcription level of *tofR*, but not that of *tofI*, was

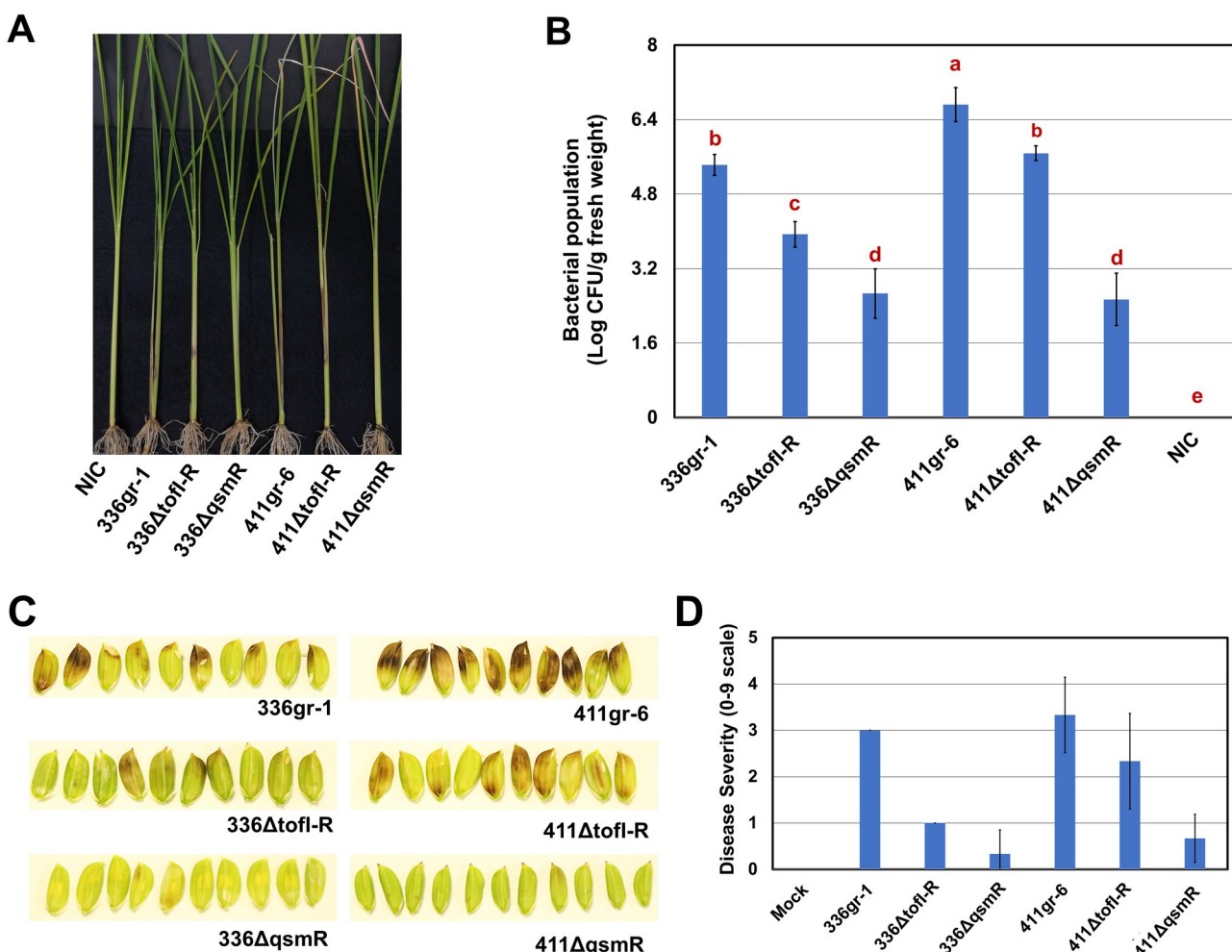

**Fig 6. The *ΔtofI-R* and *ΔqsmR* phenotypes of *Burkholderia glumae* strains in rice plants.** A) The disease symptoms on the sheaths of rice seedlings. B) The bacterial population in rice plants inoculated with the *B. glumae* strains at the seedling stage. The bacterial population was determined by real-time PCR 14 days after inoculation. Error bar represents the data of five replications, and the letters on the top of columns indicate statistically significant differences at *P* < 0.05 based on Tukey's post-hoc test. NIC means 'no inoculation control.' Two independent experiments conducted for panels A and B yielded similar results. C) The disease symptoms on rice grains caused by the *B. glumae* strains. The photo was taken 6 days after inoculation. D) The disease severity on rice panicles caused by the *B. glumae* strains. Disease severity was rated 6 days after inoculation. The experiment for panels C and D was conducted once with six replications.

significantly lower in 336ΔqsmR compared to its parent strain 336gr-1, consistent with our previous study [11]. However, in the other virulent strain 411gr-6, it was *tofI*, instead of *tofR*, that was suppressed by *qsmR* deletion (Fig 7B). Expression of *toxA* was completely suppressed in both 336ΔqsmR and 336ΔtofI-R, while it was partially (but significantly) suppressed in 411ΔtofI-R but completely suppressed in 411ΔqsmR (Fig 7B). These qRT-PCR results of the two virulent strains and their *ΔtofI-R* and *ΔqsmR* derivatives were congruent with their toxo-flavin production phenotypes shown in Figs 2 and 5. Moreover, the production of AHL signal was not abolished but reduced in 336ΔqsmR, which was consistent with the qRT-PCR data showing partial reduction of *tofR* expression in the same strain (Fig D in S1 Text). Remarkably, the avirulent strain 257sh-1 also produced the AHL signal, but its production was not reduced in the *ΔqsmR* derivative of 257sh-1, 257ΔqsmR (Fig D in S1 Text). We also observed that addition of C8-HSL did not recover the expression of *toxA* and *prtA* (the gene conferring the

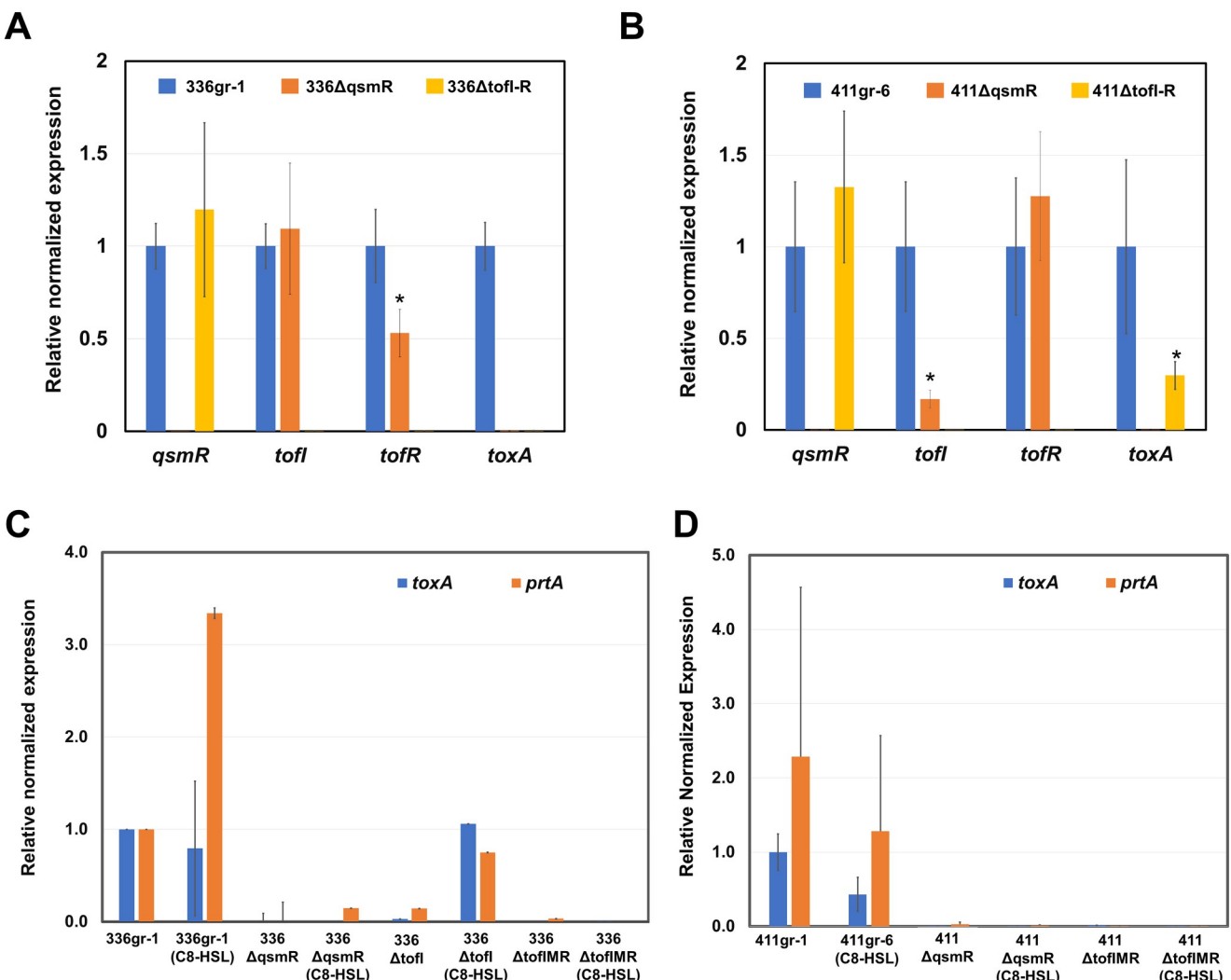

**Fig 7. Effects of *ΔqsmR* and *ΔtofI-R* mutations on the transcription of *qsmR*, *tofI*, *tofR*, and *toxA* in *Burkholderia glumae* strains 336gr-1 and 411gr-6.** A) Effects of *ΔqsmR* and *ΔtofI-R* mutations in 336gr-1. B) Effects of *ΔqsmR* and *ΔtofI-R* mutations in 411gr-6. C) Effects of *ΔqsmR* and *ΔtofI-R* mutations in 336gr-1 in the presence of 1 μM C8-HSL. D) Effects of *ΔqsmR* and *ΔtofI-R* mutations in 411gr-6 in the presence of 1 μM C8-HSL. The relative transcription levels of the *tofI*, *tofR*, *toxA*, *qsmR*, and *prtA* genes were determined by qRT-PCR. Each gene's fold change was determined using the $2^{-\Delta\Delta Ct}$ method, and the reference genes *gyrA* and 16S rRNA were used to normalize the expression level. Bars with an asterisk on the top indicate significant differences compared to the expression levels of the corresponding parent strain, 336gr-1 or 411gr-6, at $P < 0.05$ based on the two-tail t-test. Experiments for this figure were repeated twice, and similar results were obtained.

extracellular protease activity) in the *ΔqsmR* derivatives of 336gr-1 and 411gr-6 (Fig 7C and 7D). Based on these results together, we conclude that *qsmR* and TofIR QS are functionally independent, except that *qsmR* partially influences the TofIR QS system.

## The T50K substitution in QsmR caused a predicted structural change in the N-terminus and was responsible for the avirulence phenotype of 257sh-1

Based on the observed pivotal function of *qsmR* in virulence for both 336gr-1 and 411gr-6 (Figs 5–7) and the amino acid sequence variation specific to the virulence phenotype (Table C in S1 Text), we compared the putative amino acid sequences of *qsmR* between the virulent

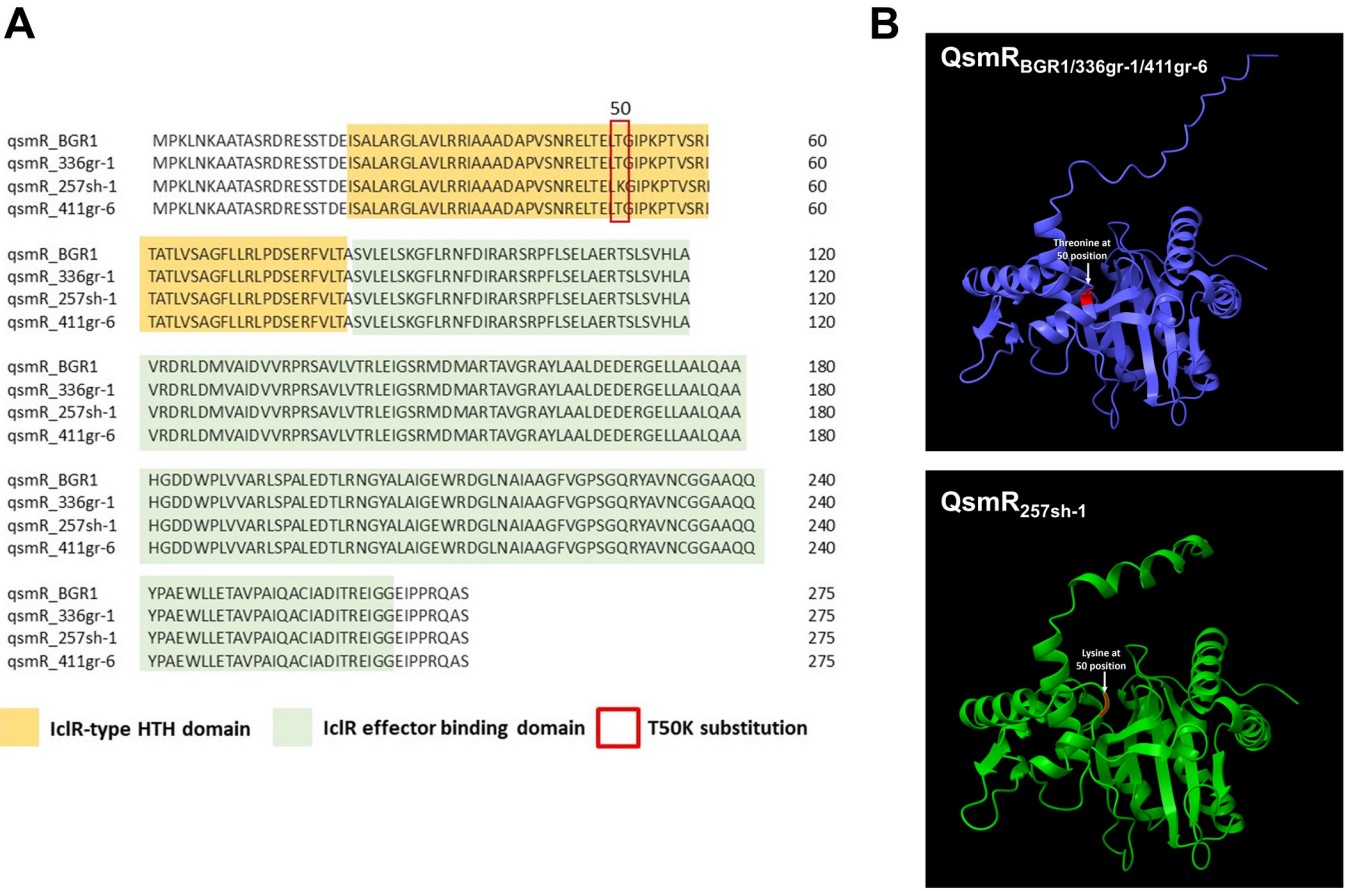

**Fig 8. Comparison of predicted amino acid sequences of the QsmR protein from different strains of *Burkholderia glumae*.** A) An alignment of QsmR sequences and two major domains of the protein predicted using MUSCLE in Unipro UGENE software [20]. The T50K substitution in 257sh-1 is indicated by a red box. B) The tertiary structures of the QsmR proteins from the virulent (336gr-1, 411gr-6, and BGR1) and avirulent (257sh-1) strains. The protein structures were predicted using Alphafold2 [21]. Visualization and silico analysis of the protein were performed using UCSF ChimeraX [22]. The highlighted area in the protein structures denotes the T50K substitution.

(336gr-1 and 411gr-1) and the naturally avirulent (257sh-1) strains. From this sequence comparison, we identified one variation in the amino acid sequence of the *qsmR* gene specific to the avirulent strain 257sh-1, which was the T50K substitution predicted in the *qsmR* coding sequence (Fig 8A). Further analysis using the AlphFold2 software predicted the effects of this variation in the protein structure and functions, in which the single amino acid variation resulted in a remarkable change in the structure at the N-terminal region (Fig 8B and E in S1 Text). Specifically, the QsmR structure of the virulent strains has only loops and turns in its N-terminal region, which becomes an alpha helix structure (from Lys 6 to Aps 19) in that of the avirulent strain A257 (Fig 8B and E in S1 Text).

Further, we examined if the T50K substitution in 257sh-1 contributes to the avirulence phenotype of 257sh-1. For this, a *qsmR* clone (p*qsmR-1*) derived from the virulent strain *B. glumae* 336gr-1 was introduced to *B. glumae* 257sh-1 and tested for its ability to restore the virulence of the avirulent strain. Remarkably, *B. glumae* 257sh-1 carrying the *qsmR* clone, p*qsmR-1*, exhibited toxoflavin production (Fig 9A) and gain of virulence in the surrogate virulence assays using onion bulb scales (Fig 9B). Another *qsmR* clone, p*qsmR-2*, which was constructed independently, also restored the virulence of 257sh-1 (Fig 9B). *B. glumae* 257sh-1 carrying

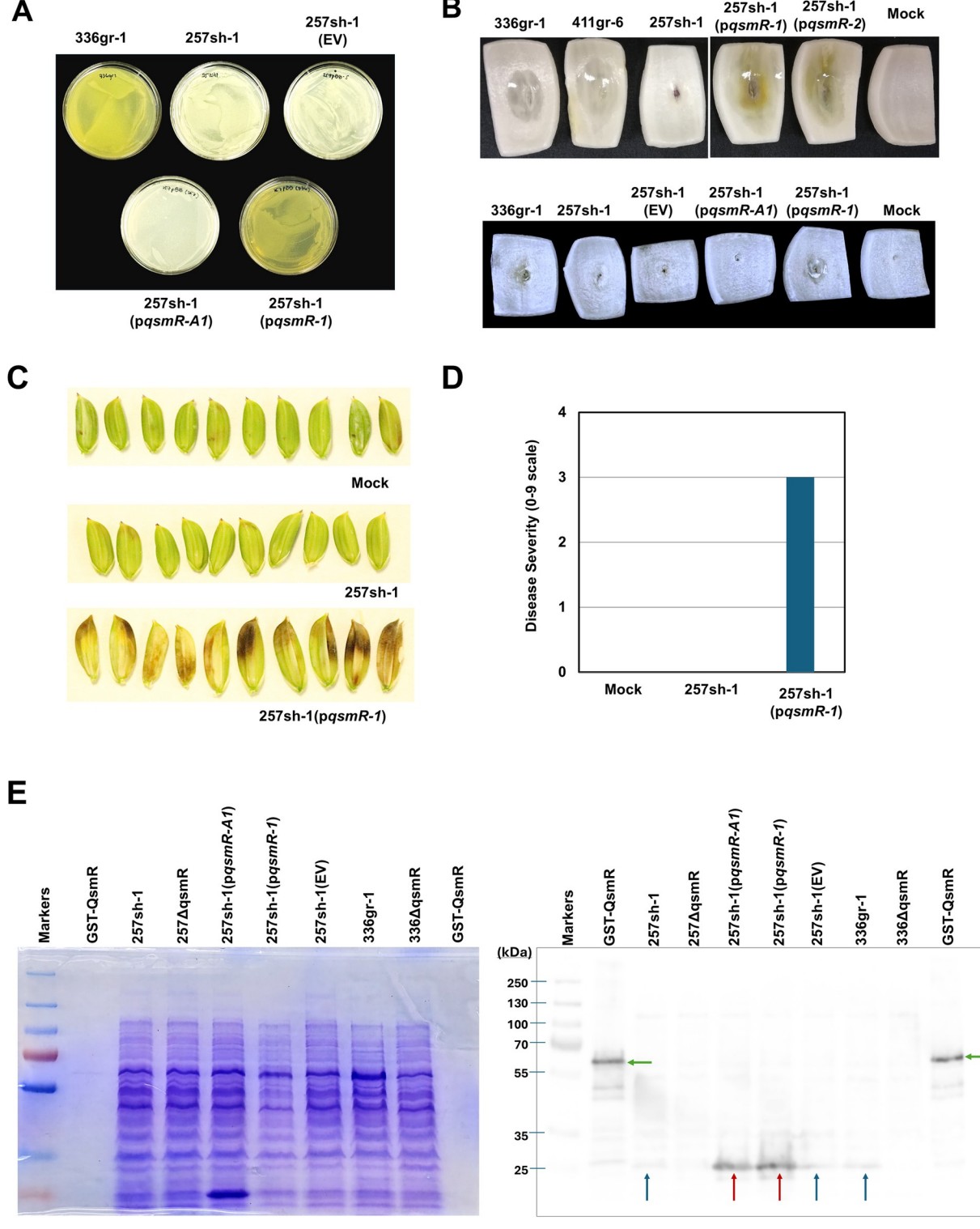

**Fig 9. Restoration of virulence in the natural avirulent strain of *B. glumae*, 257sh-1, by heterologous expression of *qsmR* from the virulent strain, 336gr-1.** A) Restoration of toxoflavin production by a *qsmR* clone carrying the virulent allele, p*qsmR-1*. A *qsmR* clone carrying the avirulent allele, p*qsmR-A1*, did not restore toxoflavin production. The photo was taken 24 h after inoculation and following incubation at 37˚C. B) Restoration of virulence in the surrogate virulence assay system using onion scales by two independent *qsmR* clones carrying the virulent allele, p*qsmR-1* and p*qsmR-2*. A *qsmR* clone carrying the avirulent allele, p*qsmR-A1*, did not restore the virulence phenotype. The photos were taken 48 h after inoculation and following incubation at 30˚C. The experiments for A) and B) were conducted more than three times,

consistently yielding similar results. C&D) Restoration of virulence in rice panicles by p*qsmR-1*. The photo and the disease severity data were obtained 6 days after inoculation. The experiment with rice plants for panels C and D was conducted once in the greenhouse with six replications. 257sh-1(EV) and 257sh-1(p*qsmR-A1*) indicate the 257sh-1 carrying the empty vector, pBBR1MCS-5, and a clone of the avirulent *qsmR* allele, respectively. 'Mock' indicates the inoculation with the buffer (10 mM MgCl$_2$) used for bacterial suspensions. E) A western blot image showing QsmR protein expression in the strains of *B. glumae*. The GST-tagged QsmR protein extracted from *E. coli* is included as the positive control. The Coomassie-stained polyacrylamide gel (left) indicates the amount of each protein sample transferred to the nitrocellulose membrane exhibiting the QsmR signal (right). The green arrows indicate the GST-tagged QsmR protein. The red and blue arrows point to the overexpressed QsmR from the *qsmR* clones and the native QsmR protein from the wild type *B. glumae* strains 336gr-1 and 257sh-1, respectively. The presented Western blot result represents three independent experiments that exhibited the same pattern.

p*qsmR-1* also showed a strong virulence in rice, like the virulence strains *B. glumae* 336gr-1 and 411gr-6 (Fig 9C and 9D). Either the same plasmid carrying *qsmR*$_{257sh-1}$, p*qsmR-A1*, or the vector itself, pBBR1MCS-5, did not restore the virulence of *B. glumae* 257sh-1, ruling out the possibility that the restoration of virulence by p*qsmR-1* or p*qsmR-2* is due to the overexpression of *qsmR* or the vector (Fig 9A and 9B). Protein expression from the native and cloned *qsmR* gene of each strain was confirmed by the Western blot analysis using anti-QsmR polyclonal rabbit antibodies (Fig 9E).

Furthermore, we made an allelic exchange of the *qsmR* locus via double homologous recombination between the 336gr-1 genome and a *qsmR*$_{257sh-1}$ clone in the suicide vector pKKSacB, and *vice versa* (between the 257sh-1 genome and a *qsmR*$_{336gr-1}$ clone in the same vector). Eight and twelve recombinant derivatives of 257sh-1 and 336gr-1, respectively, were obtained from this allelic exchange trial, and they were examined for their toxoflavin phenotypes and *qsmR* sequences. As shown in Fig 10, all three recombinants of 257sh-1 that have the A to C change at the 149[th] nucleotide position conferring T50 in QsmR gained the ability to produce toxoflavin, while those retained the A149 nucleotide sequence did not (Fig 10A). In the same way, the nine recombinants that have the C to A change at the 149[th] position conferring the T50K substitution lost the ability to produce toxoflavin, while those retained the C149 nucleotide sequence did not (Fig 10B). The other 257sh-1 specific SNP at the 270[th] position, which does not change amino acid residue, did not affect the toxoflavin phenotype of 336gr-1 (Fig 10B). Collectively, it is evident that the disabled *qsmR* function by the T50K substitution is the cause of the lost virulence of *B. glumae* 257sh-1. We also tested the phenotype of a 257ΔtofI-R complemented with p*qsmR-1*, 257ΔtofI-R(p*qsmR-1*). Interestingly, this strain regained the toxoflavin producing ability when grown on LB agar but not in LB broth (Fig F in S1 Text).

## Comparative analysis of the transcriptomes revealed genes co-regulated and differentially regulated by *qsmR* and TofIR QS

To determine the differential regulatory functions between TofIR QS and *qsmR* in *B. glumae*, transcriptomic analysis was performed by comparing the transcriptome profiles of the wild type 336gr-1 with 336ΔtofI-R and 336ΔqsmR. The principal component analysis with the RNA-seq data showed that the variations in the transcriptome profiles among the bacterial samples were nicely clustered according to their genotypes and the growth stages (the exponential phase vs. the early stationary phage), indicating that the dataset was highly reliable (Fig 11A). A total of 844 and 846 DEGs were identified from the comparisons of 336gr-1 with its Δ*qsmR* and Δ*tofI-R* derivatives, respectively, at the cutoff value of FDR 0.05 (Fig 11B and S1 Table). Among the DEGs depending on each mutant derivative, the majority were specific to Δ*qsmR* or Δ*tofI-R*, where 367 and 96 genes were up- and down-regulated only in 336ΔqsmR, and 173 and 292 genes were in 336ΔtofI-R, respectively (Fig 11B). Whereas 331 genes showed the same up or down regulation patterns between the Δ*qsmR* and Δ*tofI-R* genotypes, among which 111 and 220 genes were up- and down-regulated in both 336ΔqsmR and 336ΔtofI-R

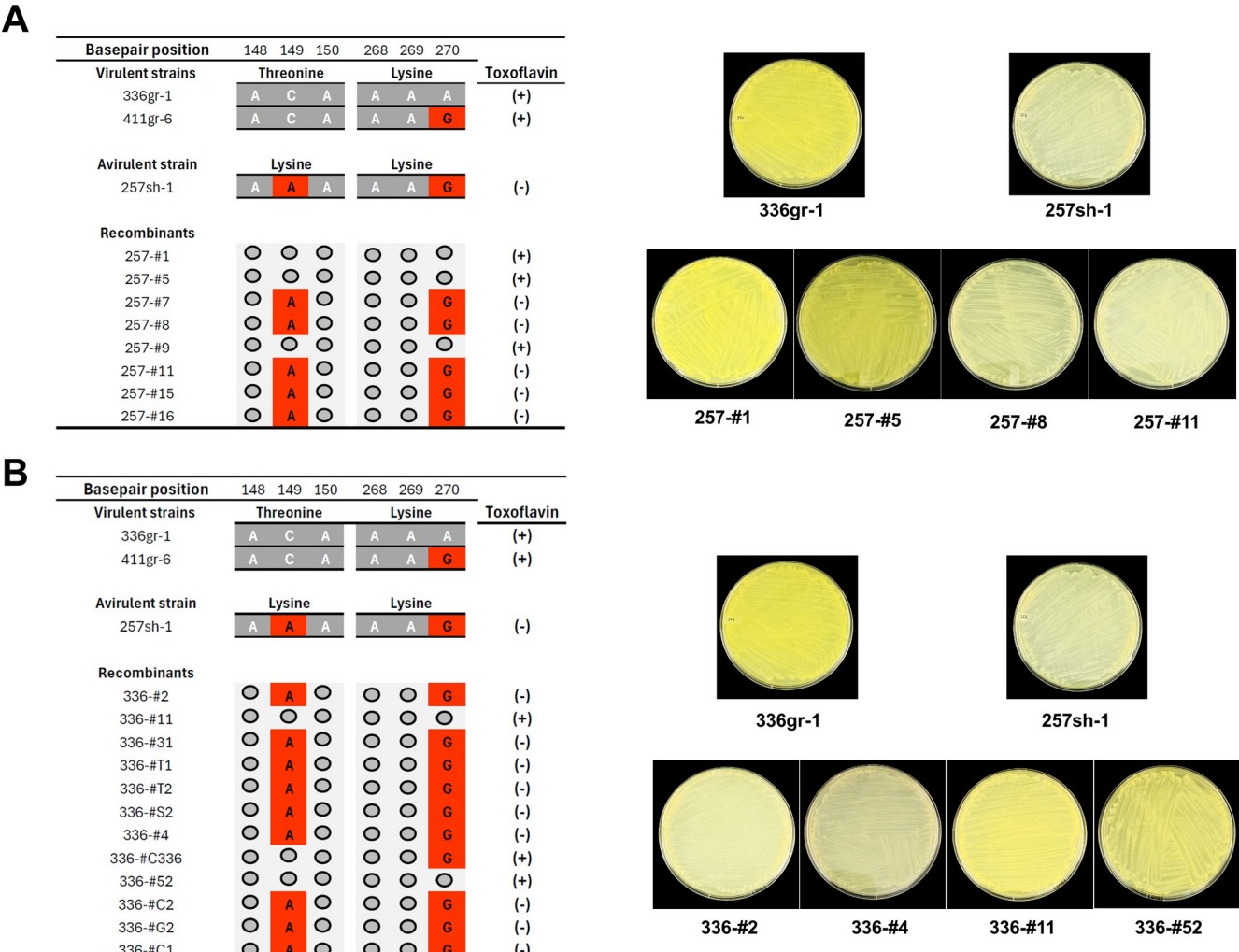

**Fig 10. Phenotypes of *B. glumae* 257sh-1 (A) and *B. glumae* 336gr-1 (B) in toxoflavin production dependent on allelic exchanges of the *qsmR* gene nucleotide sequence, C149A and A270G.** A) Derivatives of *B. glumae* 257sh-1 having the C149 allele from *B. glumae* 336gr-1 exhibit toxoflavin-positive phenotypes. B) Derivatives of *B. glumae* 336gr-1 having the A149 allele from *B. glumae* 257sh-1 exhibit toxoflavin-negative phenotypes. Experiments for this figure were conducted twice, consistently yielding similar results. Photos were taken 24 h after inoculation and following incubation at 37°C.

(Fig 11B). Besides these DEGs, 50 genes showed opposite regulation patterns between the ΔqsmR and ΔtofI-R genotypes, among which 48 genes were up-regulated in 336ΔqsmR but down-regulated in 336ΔtofI-R, and 2 genes showed the reverse pattern (Fig 11B).

Further, we examined the expression patterns of known and potential virulence-related genes in each genotype at the early stationary phase (OD$_{600}$ = 1.0) to compare the regulatory functions of TofIR QS and *qsmR* for bacterial pathogenesis. In both 336ΔqsmR and 336ΔtofI-R, genes for toxoflavin production and the PrtA metalloprotease, which is responsible for the extracellular protease activity of *B. glumae*, were down-regulated compared to the wild type 336gr-1, which conformed to their phenotypes observed in this study (Fig 11C). We also conducted the same analysis with the three wild type strains, 336gr-1, 411gr-6 and 257sh-1. However, we observed a much lower depth of differential transcriptome pattern between 336gr-1 and 257sh-1 compared to between 336gr-1 and 336ΔqsmR or 336gr-1 and 336ΔtofI-R, despite the drastic difference between the two wild type strains in pathogenicity (Fig 11C and

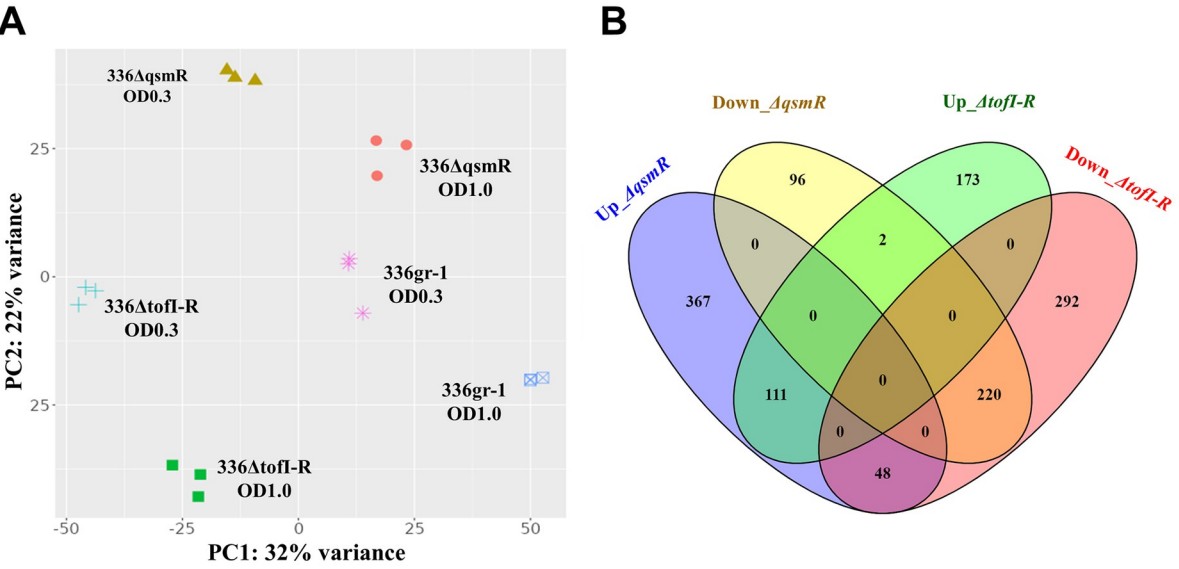

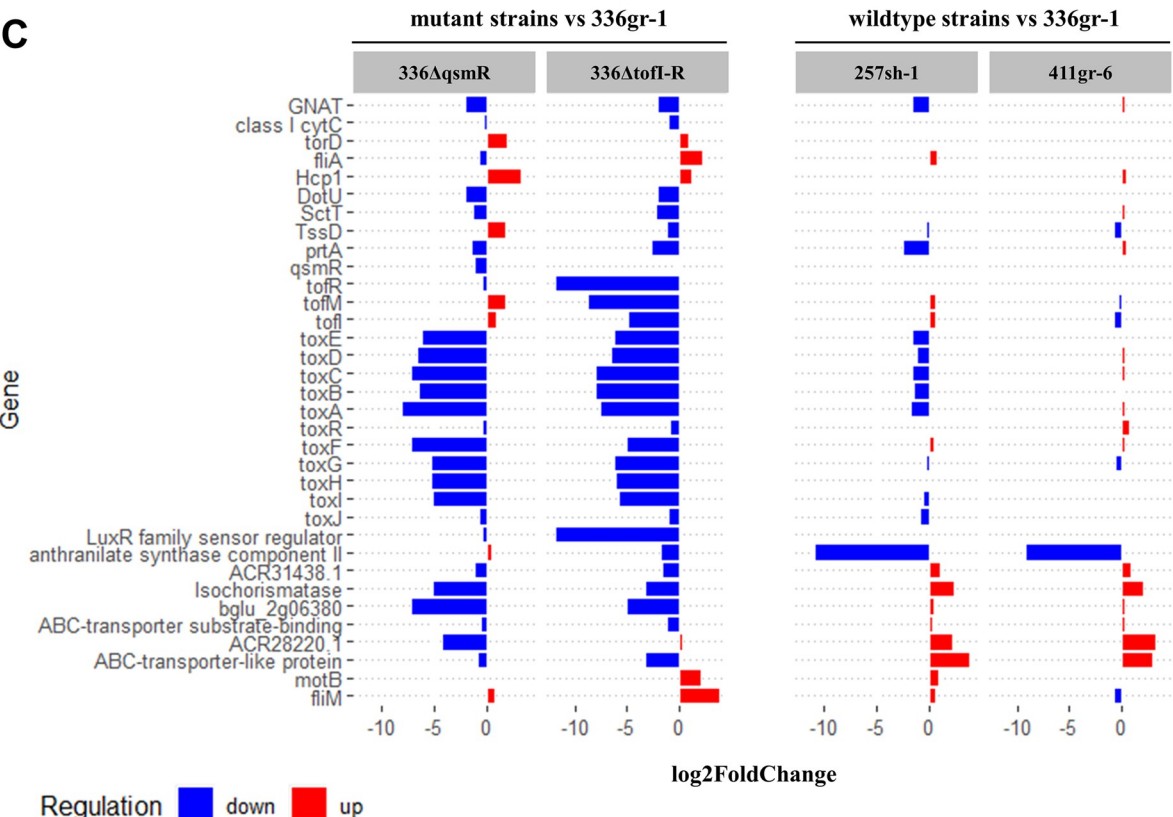

**Fig 11. The differential transcriptome profiles of the wild type and mutant strains in this study.** A) The PCA plot showing the variations of transcriptome profiles depending on the genotypes and the growth stages of *B. glumae* 336gr-1. Gene expression data was first normalized using EdgeR [23], and PCA plot was generated using the iDEP.96 platform (http://bioinformatics.sdstate.edu/idep96/). B) The Venn diagram showing the numbers of differentially regulated genes by the TofIR QS and the *qsmR* gene in *B. glumae* 336gr-1. The plot was created using the online tool, Venny 2.1.0 (https://bioinfogp.cnb.csic.es/tools/venny/), and the list of DEGs from DeSeq2 analysis was used as the input file [24]. C) Differential expression of the known and predicted virulence-related genes in the mutant and wildtype strains compared with the reference strain 336gr-1. The gene functions were annotated based on the reference genome BGR1 [25]. The horizontal bars indicate log2-transformed fold-changes of genes regulated in each strain at the early stationary phase.

S1 Table). Between the two virulent strains, 336gr-1 and 411gr-6, the depth of differential expression of virulence-related genes was minimal (Fig 11C).

## Deletion of *qsmR* also caused complete loss of toxoflavin production and extracellular protease activity in additional 20 strains of *B. glumae* tested

To examine if the differential regulatory functions of TofIR QS and *qsmR* in other strains of *B. glumae*, *ΔtofI-R* and *ΔqsmR* mutations were made with additional 20 strains of *B. glumae* isolated from the rice fields in Louisiana and adjacent states, and the individual *ΔtofI-R* and *ΔqsmR* derivatives of each strain were tested for their phenotypes in toxoflavin production and extracellular protease activity. Among the 20 strains tested, 11 and 4 strains retained the ability to produce toxoflavin and extracellular protease activity, respectively, in the *ΔtofI-R* background like 411gr-6 (Table 1). All 4 strains that were positive in protease activity were also positive in toxoflavin production. The pigmentation trait on CPG agar was not correlated with the differential phenotypes in toxoflavin and extracellular protease from the *ΔtofI-R* mutation (Table 1). Remarkably, all the 20 strains additionally tested showed complete loss of both toxoflavin production and extracellular protease activity in the *ΔqsmR* background, which was consistent with the other results for 336gr-1, 411gr-6 and 257sh-1, and confirmed the essential

**Table 1. Phenotypes of *ΔtofI-R* and *ΔqsmR* mutations observed in additional U.S. strains of *Burkholderia glumae*.**

| Strains | Origin | Pigmentation on CPG agar | *ΔtofI-R*[a] | | *ΔqsmR*[b] | |
|---|---|---|---|---|---|---|
| | | | Toxoflavin[c] | Protease[d] | Toxoflavin[c] | Protease[d] |
| 336gr-1 | Louisiana | - | - | - | - | - |
| 411gr-6 | Louisiana | + | + | + | - | - |
| 190gr-1 | Louisiana | + | - | - | - | - |
| 318gr-4 | Arkansas | - | + | + | - | - |
| 403gr-2 | Arkansas | - | + | - | - | - |
| 243gr-3 | Louisiana | + | - | - | - | - |
| 106sh-11 | Louisiana | - | + | - | - | - |
| 398gr-3 | Arkansas | - | + | + | - | - |
| 218sh-1 | Louisiana | + | - | - | - | - |
| 250sh-1 | Louisiana | - | + | - | - | - |
| 99sh-14 | Louisiana | + | - | - | - | - |
| 189gr-4 | Texas | + | - | - | - | - |
| 241gr-3 | Louisiana | + | - | - | - | - |
| 249sh-1 | Louisiana | + | - | - | - | - |
| BGLA9-1 | Louisiana | - | - | - | - | - |
| BGLA9-3 | Louisiana | + | + | - | - | - |
| BGLA13-1 | Louisiana | - | - | - | - | - |
| BGLA14-3 | Louisiana | - | + | - | - | - |
| BGLA22-1 | Louisiana | + | + | + | - | - |
| BGLA22-2 | Louisiana | + | + | - | - | - |
| BGLA22-3 | Louisiana | - | + | + | - | - |
| BGLA22-4 | Louisiana | - | + | - | - | - |

[a]: Deletion mutation of the *tofI/tofM/tofR* quorum-sensing gene cluster.

[b]: Deletion mutation of the *qsmR* gene.

[c]: Toxoflavin production based on visual observation of the yellow pigment diffused in the LB agar.

[d]: Extracellular protease activity based on visual observation of the halo zone surrounding bacterial colony on the NA agar amended with 1% skim milk.

regulatory role of *qsmR* for the virulence-related functions across the strains of *B. glumae* (Table 1).

## Discussion

In the present study, we discovered the significant variations among strains of *B. glumae* in the regulatory system for bacterial pathogenesis through a comprehensive genetic study of *B. glumae* with three strains showing differential phenotypes in physiological and pathological traits, 336gr-1 (non-pigment-producing, virulent), 411gr-6 (pigment-producing, highly virulent), and 257sh-1 (pigment-producing, naturally avirulent). Most importantly, we found that *qsmR* is a key regulatory gene for the virulence functions of *B. glumae* across all the strains tested, while the role of TofIR QS is variable among the strains. We reached this conclusion based on two lines of research outcomes from this study.

First, comparative analysis of the whole genome sequences of the three strains, 336gr-1, 411gr-6, and 257sh-1, revealed that only the *qsmR* coding sequence contained an amino acid sequence variant correlated with the virulence phenotype, which was the lysine residue at the 50th amino acid position (K50) specific to the avirulent strain 257sh-1. In the two virulent strains 336gr-1 and 411gr-6, threonine is the residue at the corresponding position (T50). To examine if this amino acid substitution confers the differential virulence phenotypes, we implemented a heterologous expression of a *qsmR* clone from the virulent strain 336gr-1, which has the T50 residue, in 257sh-1. This heterologous expression converted 257sh-1 to be virulent. Furthermore, we observed that the allelic exchanges causing T50K in 336gr-1 and K50T in 257sh-1 both resulted in the conversion of their phenotypes in toxoflavin production (Fig 10). These experimental data indicate that the T50K substitution is the cause of the avirulence phenotype of 257sh-1. Our protein structure analysis of QsmR$_{336gr-1/411gr-6}$ (T50) and QsmR$_{257sh-1}$ (K50) revealed that the T50K substitution in 257sh-1 is predicted to cause a remarkable structural change in the N-terminus part. We can rule out the possibility that the lost function of QsmR$_{257sh-1}$ might be due to a lack of protein stability, as the accumulation levels of QsmR protein were comparable between QsmR$_{336gr-1}$ and QsmR$_{257sh-1}$, regardless of whether they were expressed from native or cloned *qsmR* genes (Fig 9E).

Second, phenotype analysis of the mutant derivatives that have a deletion of the *tofI/tofM/ tofR* gene cluster (336ΔtofI-R and 411ΔtofI-R) or the *qsmR* gene (336ΔqsmR and 411 ΔqsmR) clearly indicated that *qsmR* is required for the virulence of *B. glumae* in both the virulent (336gr-1) and hypervirulent (411gr-6) strains. In contrast, the requirement of TofIR QS for virulence is substantially variable between the two virulent strains. In this study, extracellular protease activity and toxoflavin production were abolished in 336ΔtofI-R, which was consistent with previous studies that showed the dependency of toxoflavin and extracellular protease production on the TofI/TofR QS system [6,11]. However, 411ΔtofI-R retained the ability of those virulence-related functions at a similar level to the 336gr-1 wild type, although it was lower than its parent strain 411gr-6. These phenotypes of the wild type and mutant strains were highly correlated with the results of the virulence assay on rice plants, supporting the validity of the observed phenotypes. This result was intriguing because this type of TofIR QS-independent virulence-related function has not been reported by any other previous studies with various strains of *B. glumae*. In contrast, both 336ΔqsmR and 411ΔqsmR were completely deficient in both virulence-related functions, indicating the essential regulatory role of *qsmR* for bacterial pathogenesis of *B. glumae*.

It is true that the function of *qsmR* shown as the phenotypes of the *qsmR*-deficient mutants, 336ΔqsmR and 411ΔqsmR, should be validated by complementation assays through heterologous expression of *qsmR*. However, we could not observe regaining of virulence phenotype in

complementation assays, except from 257sh-1(p*qsmR*), suggesting that the feasibility of complementing *qsmR* mutants is varying among strains of *B. glumae*. We even observed that the heterologous expression of *qsmR* in the virulent strain 336gr-1 caused reduction in toxoflavin production, suggesting that proper function of *qsmR* requires a precisely regulated expression level.

To determine if other strains of *B. glumae* operate the TofIR QS-independent virulence-related functions like 411gr-6, we generated *ΔtofI-R* derivatives of 20 virulent strains isolated from the rice fields in Louisiana and adjacent states. Then we tested their ability to produce toxoflavin and extracellular protease. Interestingly, 11 out of the 20 strains tested showed TofIR QS-independent virulence-related functions like 411gr-6. Thus, the virulence-related phenotypes of the *ΔtofI-R* derivatives of 411gr-6 and other strains observed in this study disprove the 'implicit consent' that the virulence of *B. glumae* is dependent on TofIR QS. In contrast, all the 20 strains tested completely lost the toxoflavin production functions when *qsmR* was deleted (Table 1). This result also provides another line of evidence that proves the essential function of *qsmR* in the virulence of *B. glumae*.

The plasticity of the regulatory function of TofIR QS that we observed with multiple strains of *B. glumae* is reminiscent of previous studies that implied variable functions of QS genes among the subgroups of this bacterial species. We previously discovered that the single-gene mutants deficient in *tofI* or *tofR* gene of 336gr-1 retained virulence-related functions, depending on culture conditions; the mutants produced toxoflavin in the solid medium condition but not in the liquid medium condition, while they showed extracellular protease activity in both solid and liquid medium conditions [10,11]. However, this result contradicted the work performed by another research group, in which a *tofI* deletion mutant of the strain BGR1(a Korean strain) showed complete loss of toxoflavin production regardless of the medium conditions [6]. This difference in the *tofI* mutant phenotype between the strains 336gr-1 and BGR1 could be explained by a subsequent study from another research group with an extended pool of strains. In that study, 14 strains of *B. glumae* isolated from Japan were tested for their *tofI* mutant phenotypes in toxoflavin production, in which the majority of the strains (11 out of 14 strains) retained the toxin production in the absence of the functional *tofI* gene regardless of the solid or liquid medium condition, while one strain showed the *tofI⁻* phenotype like 336gr-1 (toxoflavin in the solid medium condition but not in the liquid medium condition) and the rest of them showed the phenotype like BGR1 (no toxoflavin in both conditions) [14]. These previous reports indicate that regulatory plasticity exists even in the TofIR QS system among strains of *B. glumae* species.

Meanwhile, the differential phenotypes of the TofIR QS-deficient mutants between 336gr-1 and 411gr-6 in toxoflavin production and extracellular protease activity were consistent with their virulence phenotypes in rice plants and the qRT-PCR data showing the expression level of the genes involved in toxoflavin production, which strongly supports the credibility of the mutant phenotypes determined in this study. In the virulence assays using rice plants, 411*ΔtofI-R* retained its ability to cause symptoms on rice panicles and sheaths, which was at a similar level of virulence to 336gr-1, although lower than its parent strain 411gr-6. Similarly, in the qRT-PCR assays to determine the expression level of the genes involved in toxoflavin production, the gene expression data for *toxA*, *toxR*, and *toxJ*, which are for toxoflavin biosynthesis (*toxA*) and its regulation (*toxR* and *toxJ*), were overall congruent with the observed phenotypes of each wild type or mutant strains. The highly virulent strain, 411gr-6, showed higher expression levels of those genes compared to the virulent strain 336gr-1, which were at much lower levels in the TofIR QS-deficient background of each virulent strain compared to its respective parent wild type strain. Especially, the expression level of *toxA* was retained remarkably in 411*ΔtofI-R* even though it was significantly less than in the wild type 411gr-6

(Fig 4), which was consistent with the phenotypes of 411*ΔtofI-R* in toxoflavin production and virulence and indicates the presence of TofI/TofR-independent pathway(s) to control toxoflavin biosynthesis.

The TofIR QS-deficient phenotypes of 336gr-1 and 411gr-6, as well as of 20 additional strains observed in this study, indicate that certain groups of *B. glumae* strains are less dependent on TofIR QS for virulence, implying the presence of additional regulatory/signaling pathway(s) for pathogenesis that is independent of TofIR QS. For a functional genomics study to identify genes involved in the TofIR QS-independent functions of bacterial pathogenesis, we screened a random mutant library of *B. glumae* 411ΔtofI-R constructed using mini-Tn5 transposon and obtained mutants showing reduced or abolished virulence-related functions. This functional genomics approach identified genes involved in transcription regulation, transportation, and signal transduction.

In contrast to the variable importance of TofIR QS for the virulence-related function of *B. glumae* in different strains, *qsmR* showed its essential role in the virulence-related functions in all the strains tested in this study. Consistent with the phenotypes observed *in vitro*, *qsmR* mutants of 336gr-1 and 411gr-6 did not cause any observable symptoms on rice seedlings, suggesting the essential role of *qsmR* for bacterial pathogenesis in the host plant. On the other hand, the population levels of the two mutants, 336ΔqsmR and 411ΔqsmR, were maintained at a similar basal level, although they were much lower than those of the corresponding TofIR QS-deficient derivatives, 336ΔtofI-R and 411ΔtofI-R. These TofIR QS-deficient derivatives showed significantly different population levels *in planta*, consistent with their differences in virulence-related phenotypes. Unlike the phenotypes of the TofIR QS-deficient derivatives, the *qsmR*-deficient ones did not show any significant differences between 336gr-1 and 411gr-6 in all the virulence-related phenotypes tested, including the population level *in planta*, which suggests the essential role of *qsmR* as a global master regulator for bacterial pathogenesis among diverse groups of strains within *B. glumae*. Nevertheless, we could not exclude the possible existence of an unknown subgroup of *B. glumae* that operates a *qsmR*-independent pathway for pathogenesis. Currently, characterization of the *qsmR* regulon, including the identification of QsmR-binding sites, is being performed to elucidate the functional mechanism of this master regulator.

The qRT-PCR data of this study indicated that the transcription level of *qsmR* was not affected by TofIR QS in both 336gr-1 and 411gr-6. A similar result was observed in our previous study with 336gr-1 [11]. However, these results from our studies differ from an earlier study conducted by Kim *et al.* (2007) [8], in which *qsmR* was shown to depend on the TofIR QS system for its transcription in the Korean strain BGR1. These contradictory results from different strains suggest that transcriptional regulation of *qsmR* is a strain-specific trait. Regarding this notion, it is noteworthy that *qsmR* was reported to be repressed by the AHL-dependent QS in another species of *Burkholderia*, *B. thailandensis* [26], which is an opposite regulation pattern to that of *B. glumae* BGR1 and suggests the wide range of diversity in the regulation of *qsmR* genes at both strain and species levels in the *Burkholderia* genus and related bacteria.

Although the expression of *qsmR* was not affected by TofIR QS in both 336gr-1 and 411gr-6, the qRT-PCR indicated that the expression of *tofI* and *tofR* was significantly reduced by nullifying the function of *qsmR* in 411gr-6 and 336gr-1, respectively. The same pattern of the result with 336gr-1 was observed in our previous study [11], which supports the validity of the data obtained in this study. The reduction of AHL production in 336ΔqsmR compared to its parent strain 336gr-1, along with no significant difference between 257sh-1 and 257ΔqsmR in AHL production (Fig D in S1 Text), also supports the idea that functional *qsmR* partially influence the TofIR QS system. It is remarkable that the expression of *tofI* and

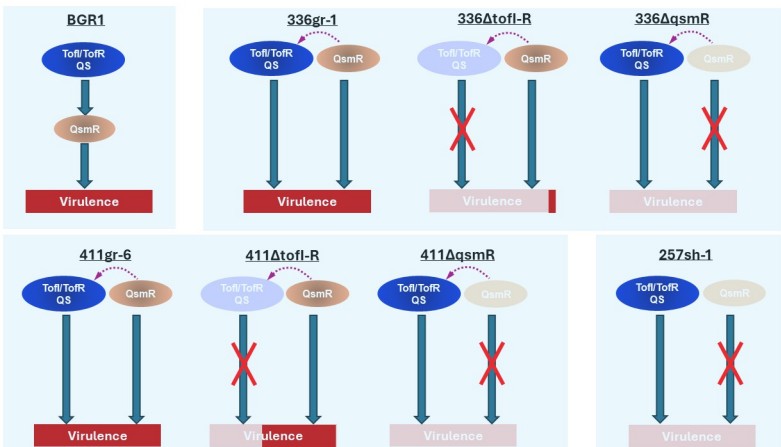

**Fig 12. A schematic working model that depicts the regulatory functions of the TofIR QS system and *qsmR*.** The dashed arrows indicate the partial influence of *qsmR* on the TofIR QS.

*tofR*, the two major genes for TofIR QS, was differentially regulated by *qsmR* in two different strains, suggesting strain-specific regulatory interactions between *qsmR* and TofIR QS. Our preliminary RNA-seq analysis to compare the regulons of *qsmR* and TofIR QS in 336gr-1 suggests that the functional relation between *qsmR* and TofIR QS in the regulation of bacterial virulence are independent of each other rather than hierarchical, because the majority of the differentially expressed genes were specific to each mutant background (*ΔqsmR* or *ΔtofI-R*).

Environmental adaptation can also play a role in the regulation of the QS regulon. Chugani *et al.* (2012) [27] tested the variation in QS regulon from strains of *Pseudomonas aeruginosa* isolated from different ecological habitats. In their findings, a set of genes under the control of QS was shared among all strains tested independent of the ecological niche. Furthermore, they also identified a strain-specific set of genes controlled by QS that correlates to the habitat of these strains tested. Variation of quorum sensing regulon among different species is often associated with the diverse habitats that bacteria occupy, and global regulatory systems, such as QS, play a crucial role during the process of adaptation. However, little information is known regarding intraspecies variation in quorum sensing regulon, especially from strains derived from the same niches such as *B. glumae* 336gr-1 and *B. glumae* 411gr-6.

*B. glumae* was considered a good model to study the QS circuit due to the presence of a single AHL-type QS regulating most virulence-related genes, making this regulatory system a good target for managing the pathogen. However, our results in this study demonstrate that the contribution of TofIR QS to the virulence of *B. glumae* is quantitatively variable among different strains. Instead, all the results gained from this study with multiple strains of *B. glumae* and their derivatives strongly support the working model that *qsmR*, rather than TofIR QS, is a universal regulatory element essential for the pathogenic functions of *B. glumae* across different subgroups within the bacterial species (Fig 12). Thus, the regulatory function of *qsmR* should be considered as a major target to manage plant diseases caused by this pathogen and related bacterial pathogens.

## Materials and methods

### Genome sequence and annotation and analysis

The complete genome sequences of the *Burkholderia glumae* strains 336gr-1, 411gr-6 and 257sh-1 were obtained and downloaded from the NCBI (https://www.ncbi.nlm.nih.gov). The annotation of the sequences was performed using Rapid Annotation using the Subsystem Technology (RAST) server (http://rast.nmpdr.org/) [28] [29]. Briefly, the annotated set of proteins of each strain was used to predict different subsystem categories using the same program based on the protein functions. Then, the obtained subsystem categories were used for further analysis and manual curation of genes associated with various secretion systems, regulatory factors, and the known and putative effectors of the *B. glumae* strains. Pairwise comparison of the amino acid sequence identity of these genes was performed using Blastp program in the National Center for Biotechnology Information (NCBI) Blast (https://blast.ncbi.nlm.nih.gov/Blast.cgi) [30]. Furthermore, known and putative type III effectors were predicted using Effectidor, an automated machine-learning-based web server to predict type-III secretion system effectors [31]. Moreover, a pairwise analysis of the average nucleotide identity (ANI) between the three strains was conducted [32]. The phylogenetic analysis of the *B. glumae* strains 336gr-1, 411gr-6, and 257sh-1 was performed to determine their relationships with other rice-associated *B. glumae* strains based on the aligned proteins and coding DNA from single-copy genes. The maximum likelihood was performed using RaxML v.8, with 1000 bootstrap replicates [33].

### Bacterial strains and growth conditions

Bacterial strains used in this study are listed in Table 2. All strains of *Escherichia coli*, *Burkholderia glumae*, and *Chromabacterium violaceum* were grown and maintained in Luria-Bertani (LB) broth or LB agar media at 30 ˚C or 37 ˚C. Antibiotics were supplemented as needed at the following concentrations: 100 µg/ml for ampicillin (Ap), 50 µg/ml for kanamycin (Km), 20 µg/ml for gentamycin (Gm), and 100 µg/ml for nitrofurantoin (Nit). LB agar plates containing 30% sucrose were used to select secondary homologous recombinants that lost the sucrose-sensitive gene, *sacB*.

### DNA cloning and amplification

Procedures for DNA cloning and amplification were conducted according to Sambrook *et al.* (2001) [38]. Genomic DNA was extracted using the GenElute Bacterial Genomic DNA Extraction kit (Sigma-Aldrich). PCR products were purified using the QuickClean 5M PCR Purification Kit (GenScript, Piscataway, NJ, USA). NanoDrop ND-100 Spectrophotometer (NanoDrop Technologies, Inc., Wilmington, DE, USA) was used to assess the quality and quantity of DNA samples. The DNA sequencing of PCR products and DNA clones was performed by Macrogen USA (Rockville, MD, USA).

### Generation of *tofI-tofR* and *qsmR* deletion mutants of 411gr-6

Since QS genes (*tofI*, *tofM*, and *tof*R) and *qsmR* are conserved in both strains of *B. glumae*, the same constructs used to generate 336ΔtofI-R [10] and 336ΔqsmR [11] were used to generate 411ΔtofI-R and 411ΔqsmR, respectively. Plasmids containing the flanking sequences of the *tofI/tofM/tofR* gene cluster (pKKSacBΔtofI-R) and the *qsmR* gene (pKKSacBΔqsmR) were transformed into *E. coli* S17-1λ*pir* through electroporation and then introduced into *B. glumae* via triparental mating with the help of *E. coli* HB101(pRK2013::Tn*7*). A single homologous recombinant of *B. glumae* was selected on an LB agar medium containing Km and Nit.

**Table 2. Bacterial strains used in this study.**

| Strains or plasmids | Properties | Reference |
|---|---|---|
| *Burkholderia glumae* | | |
| 257sh-1 | A naturally avirulent strain | [15] |
| 336gr-1 | A virulent strain | [15] |
| 411gr-6 | A highly virulent strain | [15] |
| 336ΔtofI-R | A *ΔtofI-R* derivative of *B. glumae* 336gr-1 | [10] |
| 336ΔqsmR | A *ΔqsmR* derivative of *B. glumae* 336gr-1 | [11] |
| 411ΔtofI-R | A *ΔtofI-R* derivative of *B. glumae* 411gr-6 | This work |
| 411ΔqsmR | A *ΔqsmR* derivative of *B. glumae* 411gr-6 | This work |
| *Escherichia coli* | | |
| DH10B | F *araD139* Δ(*ara, leu*)7697 *ΔlacX74 galU galK rpsL deoR* ø80d*lacZ*ΔM15 *endA1 nupG recA1 mcrA* Δ(*mrr hsdRMS mcrBC*) | [34] |
| S17-1λpir | *recA thi pro hsdR* [res- mod+][RP4::2-Tc::Mu-Km::Tn7] λ *pir* phage lysogen, Sm$^r$/Tp$^r$ | [35] |
| *Agrobacterium tumefaciens* | | |
| *A. tumefaciens* KYC55 | A biosensor strain to detect acyl-homoserine lactone signals | [50] |
| Plasmids | | |
| pKKSacB | A suicide vector; R6K γ-*ori*, RP4 *oriT*, *sacB*, Km$^R$ | [10] |
| pKKSacBΔtofI-R | A DNA construct cloned in pKKSacB for deletion of the *tofI/tofM/tofR* QS gene cluster | [10] |
| pKKSacBΔqsmR | A DNA construct cloned in pKKSacB for deletion of the *qsmR* gene | [11] |
| pKKSacBqsmR-A1 | A DNA construct cloned in pKKSacB for allelic exchange of *qsmR*$_{336gr-1}$ with *qsmR*$_{257sh-1}$ | This work |
| pSC-A-amp/kan | A blunt PCR cloning vector; f1 *ori*, pUC *ori*, *lacZ'*, Km$^R$, Amp$^R$ | Agilent Technology |
| pBBR1MCS-5 | A broad host range cloning vector, RK2 *ori*, *lacZα*, Gm$^R$ | [36] |
| pRK2013::Tn7 | A helper plasmid; ColE1 *ori* | [37] |
| p*qsmR* | A *qsmR* clone in pBBR1MCS-5, Gm$^R$ | This work |

Selected mutants were grown overnight at 30 C in LB broth without adding any antibiotics. LB agar plates containing 30% sucrose were used to select recombinants that lost sucrose-sensitive gene (*sacB*) through secondary homologous recombination. Sucrose-resistant colonies of *B. glumae* were spotted on LB agar and LB agar containing Km. Mutants that only grew in LB agar were selected for PCR confirmation. The deletion of the whole QS gene cluster (*tofI*, *tofM*, and *tofR*) was confirmed by PCR using primers TofI(H)F and TofR(H)R, correspondent to the flanking regions of *tofI* and *tofR* (Fig B in S1 Text) [10]. AHL production of the QS mutant derivatives of 336gr-1 and 411gr-6 was tested using the biosensor strain *Chromabacterium violaceum* CV026. *C. violaceum* produces a purple pigment named violacein in the presence of exogenous AHL compounds, including C6-HSL and C8-HSL [19].

## Allelic exchange of *qsmR*$_{336gr-1}$ with *qsmR*$_{257sh-1}$

The *qsmR* sequence of *B. glumae* 257sh-1 was amplified via PCR using the primers, 5'GGGTCTAGATCATGTTCGATCTGGCTGAC3' and 5'GGGATCCGTCGATTTCATCGCCAATTT3'. A 1590-bp amplicon was initially cloned in pSC-A-amp/kan, using a StrataClone PCR Cloning kit (Agilent Technologies, CA, USA). The *Bam*HI-cut fragment containing the *qsmR*$_{257sh-1}$ sequence was ligated to the suicide vector, pKKSacB, generating pKKSacBqsmR-A1. Subsequently, pKKSacBqsmR-A1 was introduced to *B. glumae* 336gr-1, and allelic exchange of *qsmR* was undergone following the same protocol for generating 411ΔtofI-R and 411ΔqsmR described in the above section. Candidate recombinants were

examined for their genotypes by sequencing the *qsmR* region using the same set of primers for *qsmR* cloning.

### Extracellular protease assay

The pectolytic activity of *B. glumae* wild type strains and mutants was first assessed on NA or LB agar plates supplemented with 1% skim milk, following Huber's method [39] with some modifications. Briefly, bacterial suspension was obtained from an overnight culture of each *B. glumae* strain or mutant in LB broth at 37 ˚C. The suspension was washed twice and resuspended in fresh LB broth to a final $OD_{600}$ of 1.0. Five microliters of the bacterial suspension were spotted on an NA or LB agar plate amended with 1% skim milk, followed by incubation at 37 C for 48 h. Extracellular protease activity was determined based on the presence of a halo zone formed around each bacterial colony due to the degradation of skim milk.

Quantification of enzymatic activity was done using azocasein (Sigma, Saint-Louis, MO, USA) as the proteolytic substrate, following Chessa's method [40]. Briefly,cell-free supernatants were obtained from an overnight bacterial culture grown in LB broth with a final $OD_{600}$ of 1.0 across samples. 100 μl of each bacterial suspension was added to 100 μl of 30 mg ml$^{-1}$ azocasein and 300 μl of 20 mM Tris/1 mM $CaCl_2$/pH 8. The reaction solutions were incubated for 1 hour at 37 C. A stop solution (500 μl of 100 mg ml$^{-1}$ trichloroacetic acid (TCA) was added to the reaction, following centrifugation at 13,000 X g for 2 min. Quantification was done by measuring the absorbance of each reaction at 366nm using a spectrophotometer (Biomate 3, Thermo Electron Corp.). A fresh medium without bacterial cells was used for the blank control.

### Toxoflavin extraction and quantification

Toxoflavin extraction was conducted following the previously established method [6] with some modifications. Briefly, bacterial strains were grown in the LB broth for 24 h at 37 C in a shaking incubator rotating at 200 rpm. One ml of bacterial supernatant was obtained after centrifugation of bacterial cultures at 16,000 *g* for 1 min. Extraction of toxoflavin from 1ml of bacterial suspension was done by mixing 1:1 ratio (v:v) of chloroform and bacterial supernatant, following centrifugation at 12,000 *g* for 5 min to separate the chloroform phase. The chloroform phase was transferred to a new microtube and placed in a fume hood overnight to evaporate. Toxoflavin was resuspended in 1 ml of 80% methanol. Absorbance was measured at 393 nm ($OD_{393}$) using the BioMate 3 spectrophotometer (Thermo Electron Corp.). Instead of a bacterial culture, a fresh medium without bacterial cells was used for the blank control.

### Quantitative reverse transcription-PCR (qRT-PCR)

RNA of *B. glumae* strains and mutants were obtained from 10 ml of overnight culture in LB broth at 37˚C. One ml of each bacterial culture was washed twice and resuspended in equal volumes of fresh LB broth. Ten μl of the resuspended bacterial cells were inoculated in 10 ml LB broth and incubated at 37˚C until bacterial culture reached $OD_{600}$ = 1.0. The LB both was amended with 1 μM of *N*-octanoyl homoserine lactone (C8-HSL) to see the effect of exogenous AHL. A 1 ml aliquot of the suspension was centrifuged to remove the supernatant, and bacterial cells were frozen in liquid nitrogen. TRIzol Reagent (Ambion Life Technologies, Grand Island, NY, USA) was used to resuspend cells. RNA extraction and DNase treatment were performed using a Direct-zol RNA MiniPrep Kit (Zymo Research, Irvine, CA, USA), following the manufacturer's instructions. cDNA was prepared using iScript gDNA Clear cDNA Synthesis Kit (Bio-Rad Laboratories, Hercules, CA, USA), following the manufacturer's instructions. Quantitative PCR (qPCR) was performed using SsoAdvance Universal SYBR- Green Supermix

(Bio-Rad Laboratories, Inc., Hercules, CA, USA), following the manufacturer's instructions. Reactions were conducted in a Bio-Rad CFX Connect thermal cycler (Bio-Rad Laboratories, Inc.). Expression values were normalized using two housekeeping genes, *gyrA* and the 16S rRNA gene. A two-tailed *t*-test was performed for the statistical analyses of the qRT-PCR data using SAS version 9.3.

## Virulence assays

The virulence assays using rice plants were conducted in the greenhouse using rice variety Trenasse which is highly susceptible to bacterial panicle blight. Rice panicles were inoculated as follows: overnight cultures of *B. glumae* strains on LB broth were washed using sterile tap water and resuspended to a final concentration of ~ $5 \times 10^8$ CFU/ml ($OD_{600} = 0.1$). Rice plants at the ~ 30% heading stage were inoculated twice with a two-day interval using a hand sprayer. Disease severity was determined based on a 0–9 scale: no symptom, 0; 1–10% symptomatic area, 1; 11–20% symptomatic area, 2; 21–30% symptomatic area, 3; 31–40% symptomatic area, 4; 41–50% symptomatic area, 5; 51–60% symptomatic area, 6; 61–70% symptomatic area, 7; 71–80% symptomatic area, 8; and more than 80% symptomatic area, 9. In each independent experiment, the disease severity of each rice plant was scored for each treatment with five or six replications.

For the virulence assays using rice seedlings, all bacterial strains were freshly grown from glycerol stocks stored at -80˚C, cross-streaked on LB agar plates, and incubated overnight at 37˚C. Bacterial cells from a single colony grown overnight on LB agar plates were collected using a toothpick and stabbed into the rice stem at the seedling stage (3-week-old).

The virulence assay using onion bulb scales was conducted following a previously established method [41]. Briefly, *B. glumae* strains and mutants were grown overnight in LB broth at 37˚C. Five μl of the bacterial suspension at $OD_{600} = 0.1$ in 10 mM $MgCl_2$ was inoculated into an onion bulb scale with a pipette. The inoculated onion scale was placed in a wet chamber at 30˚C. The virulence in the onion was determined by measuring the macerated zone around each inoculation site at 48 h after inoculation.

## Quantification of bacterial population in plants using the real time-PCR technique

The real-time PCR (RT-PCR) was conducted following the previous method established by Nandakumar *et al*. (2009) [42]. A standard curve was generated by plotting Ct values and the log values of pre-determined DNA concentrations of *B. glumae* 336gr-1 at $10^1$ to $10^8$ CFU/ml. A linear relationship was obtained between the values with a correlation coefficient ($r^2$) of 0.987. Population quantification was then determined for the *B. glumae*-infected seedlings along with healthy seedlings to exclude possible false positive data.

## Western blot analysis

*B. glumae* strains were cultured overnight (~ 18 h) in LB broth at 37˚C in a shaking incubator at 180 rpm. Two 1.5-ml aliquots of each culture were centrifuged at 10,000 rpm for 5 min using a microcentrifuge to collect bacterial cells. The pellets were then combined and resuspended in 500-μl of sterile distilled water. An equal volume of 2X modified Laemmli Buffer (4% SDS; 5% 2-mercaptoethanol; 20% glycerol; 0.004% bromophenol blue; and 0.125M Tris-HCl, pH 6.8) was added to each sample, followed by incubation in boiling water (95–100˚ C) for 10 min. Samples were centrifuged at 10,000 rpm for 10 min, and the supernatants were used for the downstream protein detection. Gel electrophoresis was performed using SDS-10% polyacrylamide gel with the Biorad system (Biorad Laboratories, Hercules, CA). Total crude protein extracts were visualized by Coomassie blue staining, and QsmR protein was detected

through Western blotting. Proteins were transferred onto the nitrocellulose membrane following the Bio-Rad protocol. The QsmR protein was probed using a QsmR-specific rabbit IgG antibody (Boster Bio, Pleasanton, CA) at a 1:500 dilution in blocking buffer (5% w/v BSA in TBST: 20mM Tris, pH 7.5; 150mM NaCl; 0.1% Tween 20). Subsequently, HRP-conjugated goat anti-rabbit IgG (Invitrogen, ThermoFisher, Waltham, MA) was applied at a 1: 20,000 dilution in 5% nonfat dry milk in TBST. The signals were visualized using SuperSignal West Pico Plus chemiluminescent substrate (Thermo Scientific) and imaged with the Azure 300 Chemiluminescent Western Blot Imager (Azure Biosystems. Dublin, CA).

## Comparative analysis of the QsmR protein structure

The structure of the QsmR protein of each strain was initially predicted based on the conserved motifs and patterns within the protein sequence using the web tool version of Prosite (a database of protein domains, families, and functional sites) (https://prosite.expasy.org). Multiple amino acid sequence alignment was performed using MUSCLE [43] in Unipro UGENE software [20]. The tertiary structures of the QsmR protein from virulent and avirulent strains of *B. glumae* were analyzed using Alphafold2 (https://github.com/sokrypton/ColabFold) [21]. This software ranks the single chains predictions based on the prediction confidence measures (pLDDT and PAE). This analysis used the predicted tertiary structure with the highest per-residue confidence (pLDDT) (scale of 0–100) score of 91.2 for further analysis. Visualization and further in silico analysis of the QsmR protein were carried out using the next-generation molecular visualization program, UCSF ChimeraX [22].

## Comparative analysis of transcriptomes

All bacterial strains were freshly grown from glycerol stocks stored at -80˚C on LB agar plates. Bacterial cells grown overnight at 37˚C were then re-suspended in LB broth and their concentrations were adjusted to $OD_{600}$ value of 0.1. A 50-μl aliquot of each normalized bacterial suspension was inoculated into 50-ml LB broth, and the cultures were incubated at 37˚C in a shaking incubator at 200 rpm. A 4-ml aliquot was sampled from each culture at the $OD_{600}$ values of 0.3 and 1.0, which represented the exponential and the early stationary phase of bacterial growth stage, respectively. Bacterial cells were collected via centrifugation and then snap-frozen with liquid nitrogen. Frozen bacterial cells were lysed by gently mixing with 1.0 ml of TRIzol (Thermo Fisher Scientific) and the total RNA was extracted from each sample using the Direct-zol RNA MiniPrep Kit (Zymo Research, Irvine, CA). All total RNA from three biological replications of each sample were sent to the Oklahoma Medical Research Foundation Genomics Core Laboratory for the library preparation and RNA sequencing. RNA libraries were prepared using a Swift Rapid RNA Library Prep kit (IDT, Carolville, IA, USA), and ribosomal RNAs were removed using an Illumina Ribo-zero rRNA depletion kit (Illumina, Inc., San Diego, CA, USA). Quality checking and normalization were conducted using a Kapa Library Quantification kit (Roche, Hague Rd., IN, USA) and the Agilent Tapestation Systems (Agilent, Santa Clara, CA, USA). Finally, libraries were sequenced on a S4 flow cell lane (2 x 150 cycle) of the Illumina NovaSeq machine.

The raw sequencing reads were pre-processed to remove adapter sequences, short reads, and low-quality sequences using the Trimmomatic tool [44], and the quality of reads was then evaluated using the FastQC tool [45]. High-quality reads were mapped to the genome of *B. glumae* BGR1 (ASM2264v1, http://www.ncbi.nlm.nih.gov/assembly/; Lim et al., 2009) [25], using the Burrows-Wheeler Aligner tool, and the featureCounts tool was used to determine the levels of gene expression [46,47]. The association of samples was assessed by conducting hierarchical clustering and principal component analyses. Briefly, genes with minimum counts per million

of 0.5 in at least two libraries were kept for analysis. Count data transformation was performed using EdgeR's log2(CPM + 4), while gene median was used for missing values imputation. The differential gene expression analysis was carried out using Deseq2 with the following threshold —FDR of 0.05 and a fold change of 2, to call a significant differentially expressed gene (DEG). Fold-change values from previous analyses were used for gene set enrichment analysis to compare the regulons in between samples using the KEGG database (https://www.genome.jp/) and the clusterProfiler package in R [48].

## AHL quantification

Production of AHLs was determined following the methods of Iqbal *et al*. [49] with some modifications. The strain *A. tumefaciens* KYC55 [50] was used as a $\beta$-galactosidase-based biosensor, which responds to the presence of AHLs by expressing *lacZ*. The overnight grown bacterial cultures were adjusted to $OD_{600} = 0.1$ and the culture supernatants were collected by centrifugation. Ethyl acetate was then added to the collected supernatants at 1:1 (v/v) ratio in 2mL microcentrifuge tubes to extract the AHL. The upper layer of the suspension was collected and airdried overnight in a fume hood then resuspended in a 1% volume of sterile distilled deionized water. Ten μL of the AHL extract from each strain was added to culture tubes containing 5 ml AT broth supplemented with 40 μg/ml *X*-gal and inoculated with the overnight grown *A. tumefaciens* KYC55 ($OD_{600} = 0.2$). The samples were incubated at 30˚C for 24 h. Control tubes were supplemented with 0 or 5 μM C8-HSL. The development of blue color was read at 635 nm. The absorbance of the negative control was subtracted from the absorbance of each sample.

## Supporting information

**S1 Text.** Table A. Genome features and statistics of the three *Burkholderia glumae* strains used in this study. Table B. Pairwise comparison of the average nucleotide identity (ANI) between *B. glumae* strains. Table C. Pairwise comparison of virulence-related gene products in the amino acid sequence identity between *B. glumae* strains. Fig A. Phylogenetic analysis of selected rice-associated *B. glumae* strains based on the aligned proteins and coding DNA from single-copy genes using RaxML v.8 program. Fig B. A schematic depiction of the *tofI/tofM/tofR* QS gene cluster deleted in the *ΔtofI-R* derivatives of 336gr-1 and 411gr-6. Black arrows indicate the primer set to confirm the gene deletion by PCR (the agarose gel image on right side). Fig C. The production of the purple pigment, violacein, by the biosensor *Chromobacterium violaceum* CV026 (McClean *et al*. 1997)[19] dependent on the acyl homoserine lactone (AHL) QS signal molecules from each strain of *B. glumae*. Fig D. The production of the AHL QS signal molecule of *B. glumae* by the *ΔqsmR* and *ΔtofI-R* or *ΔtofI* derivatives of 336gr-1 and 257sh-1. Fig E. Comparison of the structures of the QsmR proteins between the virulent strains (336gr-1, 411gr-6, and BGR1, purple) and the avirulent strain (257sh-1, green). Fig F. Restoration of toxoflavin production in 257ΔtofI-R by a *qsmR* clone carrying the virulent allele, p*qsmR-1*. The photo was taken 24 h after inoculation and following incubation at 37˚C. (DOCX)

**S1 Table. Lists of differentially expressed genes identified from the transcriptome analysis of this study.**
(XLTX)

## Author Contributions

**Conceptualization:** Jong Hyun Ham.

**Data curation:** Tiago De Paula Lelis, Jobelle Bruno, Jonas Padilla.

**Formal analysis:** Tiago De Paula Lelis, Jonas Padilla.

**Funding acquisition:** Jong Hyun Ham.

**Investigation:** Tiago De Paula Lelis, Jobelle Bruno, Jonas Padilla, Inderjit Barphagha, John Ontoy.

**Methodology:** Jobelle Bruno, Jonas Padilla, Inderjit Barphagha, John Ontoy, Jong Hyun Ham.

**Project administration:** Jong Hyun Ham.

**Resources:** Inderjit Barphagha, Jong Hyun Ham.

**Software:** Jobelle Bruno, Jonas Padilla.

**Supervision:** Jong Hyun Ham.

**Validation:** Jobelle Bruno.

**Visualization:** Tiago De Paula Lelis, Jobelle Bruno, Jonas Padilla, Jong Hyun Ham.

**Writing – original draft:** Tiago De Paula Lelis, Jobelle Bruno, Jonas Padilla, Jong Hyun Ham.

**Writing – review & editing:** Tiago De Paula Lelis, Jobelle Bruno, Jonas Padilla, John Ontoy, Jong Hyun Ham.

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
