## [Decision Letter · Decision Letter 0]

6 Jan 2024

Dear Dr. Ham,

Thank you very much for submitting your manuscript "qsmR encoding an IclR-family transcriptional factor is a core pathogenic determinant of Burkholderia glumae beyond the acyl-homoserine lactone-mediated quorum-sensing system" for consideration at PLOS Pathogens. As with all papers reviewed by the journal, your manuscript was reviewed by members of the editorial board and by several independent reviewers. In light of the reviews (below this email), we would like to invite the resubmission of a significantly-revised version that takes into account the reviewers' comments.

Your manuscript was reviewed by two experts in bacterial genetics and quorum-sensing. Both reviewers acknowledged the significant progress and the amount of work in this study. However, both reviewers have identified major issues in experiments and interpretation. The reviewers have suggested additional experiments. These experiments are required for further consideration.

We cannot make any decision about publication until we have seen the revised manuscript and your response to the reviewers' comments. Your revised manuscript is also likely to be sent to reviewers for further evaluation.

Sincerely,

Nian Wang

Academic Editor

PLOS Pathogens

Shou-Wei Ding

Section Editor

PLOS Pathogens

Kasturi Haldar

Editor-in-Chief

PLOS Pathogens

orcid.org/0000-0001-5065-158X

Michael Malim

Editor-in-Chief

PLOS Pathogens

orcid.org/0000-0002-7699-2064

Your manuscript was reviewed by two experts in bacterial genetics and quorum-sensing. Both reviewers acknowledged the significant progress and the amount of work in this study. However, both reviewers have identified major issues in experiments and interpretation. The reviewers have suggested additional experiments. These experiments are required for further consideration.

Reviewer's Responses to Questions

**Part I - Summary**

Reviewer #1: In the manuscript titled "qsmR encoding an IclR-family transcriptional factor…," Lelis et al. conducted an extensive investigation into the role of the tof QS system and the qsmR regulator in the virulence of various Burkholderia glumae clones with distinct pathogenicity profiles. The authors found that the contribution of the tof system to virulence is strain-dependent, whereas qsmR is essential for pathogenesis across all tested strains. This reinforces prior findings by the same research group, solidifying qsmR as a key virulence regulator in B. glumae.

Moreover, the authors identified a single nucleotide polymorphism (SNP) in qsmR of the non-virulent strain B. glumae 257. Through complementation analyses, they demonstrated that the introduction of qsmR from other strains via plasmid successfully restored the pathogenicity of 257. This suggests that variations in qsmR may be a contributing factor to the loss of virulence in strain 257.

This study represents a substantial effort, and the results have the potential to significantly advance our understanding of B. glumae pathogenesis. In addition, the implications of SNP variants in qsmR for the pathogenicity of the bacterial population are particularly noteworthy, holding broader significance in the fields of pathogenomics and virulence regulation.

However, I have concerns regarding the study's methodology. Many experiments lack adequate controls, and certain analyses were inaccurately described (outlined in the "major revisions" section). These issues contribute to my reservations about the persuasiveness of the results. Consequently, I do not recommend the acceptance of this manuscript.

Reviewer #2: The manuscript PPATHOGENS-D-23-02079 is well-written and shows interesting results regarding mechanism of regulation of the pathogenicity on different Burkholderia glumae strains. The authors have demonstrated that the transcription regulator QsmR is a key factor on B. glumae virulence regulation since a residue mutation (T50K) in this protein has impaired the virulence. The observed results have indicated that QsmR can differently regulate the quorum-sensing genes tofI and tofR, which might dependent on the B. glumae strain analyzed. Though deletion of qsmR promoted the lost of HSL-C8 production on mutant strains, the mechanism which qsmR regulates the virulence in B. glumae has been not determined in this work. Following I have listed some points that may help to improve further some parts of this manuscript.

**Part II – Major Issues: Key Experiments Required for Acceptance**

Reviewer #1: - The most exciting claim that the authors make in this study is that natural small variation in the coding sequence of a virulence regulator is the driving factor that dictates the aggressiveness of a pathogen. From an evolutionary standpoint, this result might have a major impact on the field of bacterial virulence regulation. However, I don't think that the authors did a good job of proving this point, and their key experiment lacked necessary controls. To address this issue, I suggest the authors consider one of the following options:

1. The authors used pBBR1MCS5 for "complementing" the 257 clone with 411/336qsmR and showed that the reintroduction restores virulence.

pBBR1MCS5 is a medium copy number plasmid (I believe it has ~20 copies), and its MCS is found after a lac promoter. Therefore, pBBR1MCS5-based complementation is a form of overexpression. With that in mind, the result observed by the authors in Fig. 9 can be caused by the overexpression of QsmR and not the variation in protein sequence. To address this, I suggest the authors clone qsmR from 257 into pBBR1MCS5, introduce it to 257, and repeat the experiment. I also strongly suggest adding a protein tag and confirming protein accumulation via Western blot. If introduction of qsmR257 does not turn 257 into a virulent strain, it will support the authors' hypothesis.

2. A more elegant way to support the hypothesis is replacing the qsmR variant in the genome of 257 with qsmR from 411/336. This will address any issues that might arise from introducing a plasmid and potential overexpression since the gene will be expressed under its natural promoter, from its original genomic location, and in a single copy. I don't have experience with Burkholderia glumae, but as far as I understand, the sacB counter-selection system works well in this bacteria, and therefore, altering qsmR in the genome is doable.

- I disagree with the interpretation of the authors regarding the differential role of the tof system in the virulence of 411 and 336. In both cases, the tof system significantly contributes in a quantitative manner, but it is not essential, unlike qsmr (which appears to have a binary effect according to the study). 411 is more pathogenic and produces more toxoflavin and exopeptidase activity than 336. Therefore, the reduction in virulence caused by the disruption of tof is more apparent in 336. I actually think that this experiment supports that the differences in the virulence of these strains are unrelated to the tof system.

- Outside of Fig. 9, the authors did not complement any of the mutants during the various analyses. Complementation is a crucial control in bacterial genetics-based studies and should be added to the experiments described in Fig. 2-7 (or at least some of them).

Reviewer #2: Major points:

1-In the text is mentioned that:“the transcription level of tofR, but not that of tofI, was significantly lower in 336ΔqsmR compared to its parent strain 336gr-1. However, in the other virulent strain 411gr-6, it was tofI, instead of tofR, that was suppressed by qsmR deletion (Figure 7B). Expression of toxA was completely suppressed in both 336ΔqsmR and 336ΔtofI-R, while it was partially (but significantly) suppressed in 411ΔtofI-R but completely suppressed in 411ΔqsmR (Figure 7B).” To strengthen these findings, I suggest to add exogenous C8-HSL in 411gr-6qmsR mutant strain culture and verify if it should recover toxA expression. It might indicate that qmsR acts regulating some point in C8-HSL biosynthesis in B. glumae strains.

2- One important point that should be determined in this work is the action mechanism of the transcription regulator qmsR on virulence regulation of B. glumae. Based on gene expression analysis the authors should verify putative QmsR-binding site sequences and run some experiments with gene reporter fused to candidate promoter sequences and/or test the binding of the putative QmsR-binding site sequences and QmsR protein with gel shift assay. Is QmsR-binding site element present in different promoters TofR (for 336gr-1) and TofI (for 411gr-6) or in other factors that act in trans to regulate the virulence in B. glumae? Can QmsR-binding site be found on promoters of T3SS-coding genes or other essential genes for virulence in B. glumae?

3- It should be depicted a model showing the interplay of QsmR and QS regulon on regulation of virulence factors of B. glumae strains and the points that are still undetermined indicated in dashed lines.

**Part III – Minor Issues: Editorial and Data Presentation Modifications**

Reviewer #1: - The pages and lines are not numbered. This made extremely difficult to refer to specific points in the text while writing the review. Please have this issue corrected during resubmission.

- Fig. 4: As far as I understand, all data presented here was normalized to wt 336, including the 411 tof mutant. A mutant should be compared to the wt of its parent strain and not to an unrelated strain. I suggest that the analysis should be split into three parts. In the first analysis, the authors should compare gene expression of 336 vs 411. In the second analysis, the tof mutant of 336 should be compared to wt 336. In the third analysis, the tof mutant of 411 should be compared to wt 411.

- Why were the experiments conducted in Fig. 2-7 done only on 411 and 336 and not on 257? Please explain.

- The transcriptomic analyses described in Fig. 10 for 336 seem detached from the study. I suggest the authors remove this data and publish it separately. The transcriptomic data can easily be presented in an entire standalone manuscript. Regardless, the RNA-seq data should be supplemented with an Excel sheet describing all the DEGs and fold changes, and raw data should be deposited in a public database.

- Fig S1: Considering that a draft genome sequences are available, why did the authors opt to use a single-copy gene for the phylogenetic analysis?

Reviewer #2: Minor points:

-Please, clarify “lost the function” in this sentence in abstract: all the virulent strains tested lost the function almost completely upon deletion of the qsmR gene.

-Page 10 “B. glumae 411gr-6 is a highly virulent strain that produces pigments on CPG medium, which have antimicrobial activities.” It should be highlighted in the introduction section in the MS which virulence factors in B. glumae are essential for pathogenicity. For example, protease production, T2SS, T3SS and effectors.

-Page 12 - PidS/pidR two component regulatory. Please, change “p”idR to capital letter

PLOS authors have the option to publish the peer review history of their article (what does this mean?). If published, this will include your full peer review and any attached files.

Reviewer #1: No

Reviewer #2: No
---

## [Decision Letter · Decision Letter 1]

23 Jun 2024

Dear Dr. Ham,

Thank you very much for submitting your manuscript "qsmR encoding an IclR-family transcriptional factor is a core pathogenic determinant of Burkholderia glumae beyond the acyl-homoserine lactone-mediated quorum-sensing system" for consideration at PLOS Pathogens. As with all papers reviewed by the journal, your manuscript was reviewed by members of the editorial board and by several independent reviewers. The reviewers appreciated the attention to an important topic. Based on the reviews, we are likely to accept this manuscript for publication, providing that you modify the manuscript according to the review recommendations.

The manuscript was sent back to the two colleagues who have reviewed the initial submission. Both reviewers commended the revision. There are only few points that need further attention. Regarding the RNA-seq data, I think it is ok to keep the data in this manuscript.

Sincerely,

Nian Wang

Academic Editor

PLOS Pathogens

Shou-Wei Ding

Section Editor

PLOS Pathogens

Michael Malim

Editor-in-Chief

PLOS Pathogens

orcid.org/0000-0002-7699-2064

The manuscript was sent back two colleagues who have reviewed the initial submission. Both reviewers commended the revision. There are only few points that need further attention. Regarding the RNA-seq data, I think it is ok to keep the data in this manuscript.

Reviewer Comments (if any, and for reference):

Reviewer's Responses to Questions

**Part I - Summary**

Reviewer #1: The current submission by Lelis et al. is a revised version of the original manuscript addressing the reviewers' comments. The authors have addressed most of the issues raised by myself in the original submission.

Although the authors failed to complement the qsmR mutant, they have convinced me that qsmR is indeed a major virulence regulator by demonstrating that this phenotype occurs in multiple independently disrupted qsmR B. glumae strains. I also agree that the expression levels of transcriptional regulators like QsmR may require fine-tuning for proper function, which could explain why complementation failed.

I still think that the gene expression analysis in Figure 4 should be separated into several subfigures, each depicting the gene expression pattern in each bacterial strain. I also think that the protein expression of QsmR257 and QsmR336 should be confirmed in 257sh-1 to support the newly depicted data in Figure 9.

Reviewer #2: The authors have addressed most of the queries in revised manuscript and added changes in text which helped to clarify some of the points indicated by the reviewers. I really miss the molecular mechanism by which QsmR can contribute in regulating the virulence factors in Burkholderia glumae, though I suppose that it should be very complex mechanism and may related with fine-tuning expression of qsmR itself and of the factors under its regulation. The findings described in the work are interesting and provoke for searching a better understanding of QmsR regulation in future researches. The findings described in the work are interesting and may help to enforce food security by chasing a inhibitor of QsmR. I recommend this revised work for publication.

**Part II – Major Issues: Key Experiments Required for Acceptance**

Reviewer #1: (No Response)

Reviewer #2: (No Response)

**Part III – Minor Issues: Editorial and Data Presentation Modifications**

Reviewer #1: 1. Figure 4: As I mentioned in the first revision cycle, I think that each mutant should be compared to its parent strain and not collectively compared to strain 336. Please reanalyze the data as requested previously and split the figure into three subfigures.

2. Figure 9: I think that the newly depicted data in Figures 9A and 9B support that variation in QsmR is a significant contributing factor to the variation in virulence identified between the strains. However, I think this data would be more convincing if complementation is supported by a western blot. If the authors demonstrate that, despite containing the plasmid-encoded qsmR variants, there is a clear difference in the protein accumulation between QsmR257 and QsmR336, the data would still be valid and could be a point for discussion since sequence variation might affect protein stability or RNA structure.

3. Table 1: Can you add another "WT" column in addition to ΔtofI and ΔqsmR?

4. I see according to the figure legends that while most experiments were repeated at least twice, some of the rice inoculation experiments were repeated only once (Figures 6C, 6D, 9C, and 9D). Is this indeed the case? If so, what was the reason that the authors did not repeat these particular experiments?

5. Regarding the incorporation of the RNA seq analysis, I suggested doing so because I feel that the authors would benefit if these results were more highlighted in a standalone independent manuscript that would place more emphasis on the transcriptome data. However, the authors can choose to keep the data as part of the manuscript if they see fit.

Reviewer #2: (No Response)

PLOS authors have the option to publish the peer review history of their article (what does this mean?). If published, this will include your full peer review and any attached files.

Reviewer #1: No

Reviewer #2: No

Figure Files:

Data Requirements:

Reproducibility:

References:

---

## [Editor Report · Decision Letter 2]

20 Sep 2024

Dear Dr. Ham,

We are pleased to inform you that your manuscript 'qsmR encoding an IclR-family transcriptional factor is a core pathogenic determinant of Burkholderia glumae beyond the acyl-homoserine lactone-mediated quorum-sensing system' has been provisionally accepted for publication in PLOS Pathogens.

Best regards,

Nian Wang

Academic Editor

PLOS Pathogens

Shou-Wei Ding

Section Editor

PLOS Pathogens

Michael Malim

Editor-in-Chief

PLOS Pathogens

orcid.org/0000-0002-7699-2064
---

## [Editor Report · Acceptance letter]

30 Sep 2024

Dear Dr. Ham,

We are delighted to inform you that your manuscript, "qsmR encoding an IclR-family transcriptional factor is a core pathogenic determinant of Burkholderia glumae beyond the acyl-homoserine lactone-mediated quorum-sensing system," has been formally accepted for publication in PLOS Pathogens.

Best regards,

Michael Malim

Editor-in-Chief

PLOS Pathogens

orcid.org/0000-0002-7699-2064